# Learning Imperfect Information Extensive-form Games with Last-iterate Convergence under Bandit Feedback

Canzhe Zhao [1]   Yutian Cheng [1]   Jing Dong [2]   Baoxiang Wang [2]   Shuai Li [1]

## Abstract

We investigate learning approximate Nash equilibrium (NE) policy profiles in two-player zero-sum imperfect information extensive-form games (IIEFGs) with last-iterate convergence guarantees. Existing algorithms either rely on full-information feedback or provide only asymptotic convergence rates. In contrast, we focus on the bandit feedback setting, where players receive feedback solely from the rewards associated with the experienced information set and action pairs in each episode. Our proposed algorithm employs a negentropy regularizer weighted by a "virtual transition" over the information set-action space to facilitate an efficient approximate policy update. Through a carefully designed virtual transition and leveraging the entropy regularization technique, we demonstrate finite-time last-iterate convergence to the NE with a rate of $\widetilde{\mathcal{O}}(k^{-1/8})$ under bandit feedback in each episode $k$. Empirical evaluations across various IIEFG instances show its competitive performance compared to baseline methods.

## 1. Introduction

In imperfect information games (IIGs), players operate with limited visibility of the game's true state, necessitating strategic decision-making based on incomplete information. Notably, the concept of imperfect-information extensive-form games (IIEFGs), as introduced by Kuhn (1953), encapsulates both the intricacies of imperfect information and the sequential nature of players' moves. This framework aptly represents a broad spectrum of real-world scenarios, such as Poker (Heinrich et al., 2015; Moravčík et al., 2017; Brown & Sandholm, 2018), Bridge (Tian et al., 2020), Scotland

[1]John Hopcroft Center for Computer Science, Shanghai Jiao Tong University [2]School of Data Science, The Chinese University of Hong Kong, Shenzhen. Correspondence to: Shuai Li <shuaili8@sjtu.edu.cn>.

*Proceedings of the $42^{nd}$ International Conference on Machine Learning*, Vancouver, Canada. PMLR 267, 2025. Copyright 2025 by the author(s).

Yard (Schmid et al., 2021), and Mahjong (Li et al., 2020; Kurita & Hoki, 2021; Fu et al., 2022). Extensive research has been devoted to identifying the (approximate) Nash equilibrium (NE) (Nash Jr, 1950) within IIEFGs. With the full knowledge of the game, various methodologies have been employed to tackle these games. These include linear programming approaches (Koller & Megiddo, 1992; Von Stengel, 1996; Koller et al., 1996), first-order optimization techniques (Hoda et al., 2010; Kroer et al., 2015; 2018; Munos et al., 2020; Lee et al., 2021; Liu et al., 2022), and counterfactual regret minimization (CFR) algorithms (Zinkevich et al., 2007; Lanctot et al., 2009; Johanson et al., 2012; Tammelin, 2014; Schmid et al., 2019; Burch et al., 2019; Liu et al., 2022).

In practical scenarios, IIEFGs might involve unknown reward distributions over information set and action spaces, thwarting the application of the above approaches for *computing* the NE in IIEFGs. In this realm, the NE in IIEFGs is typically *learned* from random samples gathered through iterative playthroughs of the game, by Monte-Carlo CFR methods (Lanctot et al., 2009; Farina et al., 2020; Farina & Sandholm, 2021), online mirror descent (OMD) or follow-the-regularized-leader (FTRL) frameworks (Farina et al., 2021; Kozuno et al., 2021; Bai et al., 2022; Fiegel et al., 2023). Notably, Bai et al. (2022) devise an OMD-based approach incorporating "balanced exploration policies" to learn an $\varepsilon$-approximate NE with a sample complexity of $\widetilde{\mathcal{O}}\left(H^3(XA + YB)/\varepsilon^2\right)$, where $H$ is the horizon length, $X, Y$ are the sizes of the information set space for the max- and min-player, and $A, B$ are the sizes of the action space for the max- and min-player. This upper bound is information-theoretically optimal with respect to all parameters except $H$, up to logarithmic factors. Building upon Bai et al. (2022), Fiegel et al. (2023) make further strides, refining the upper bound to $\widetilde{\mathcal{O}}\left(H(XA + YB)/\varepsilon^2\right)$ by harnessing FTRL with "balanced transitions", achieving (nearly) optimal sample complexity in all parameters.

Despite the (nearly) optimal leaning of the $\varepsilon$-NE in IIEFGs by Bai et al. (2022) and Fiegel et al. (2023), the algorithms in these works require to average all the policies generated during the running of the algorithms, so as to obtain the final policy profile with $\varepsilon$-NE guarantee. This is typically

termed as the *average-iterate convergence*. However, in cases when the policies in the games are approximated by nonlinear function approximation (*e.g.*, neural networks), which has achieved great empirical success in recent years (Moravčík et al., 2017; Brown & Sandholm, 2018), computing the averaged policy might even be infeasible due to the nonlinearity of such function approximations. This motivates the studies of the learning algorithms with the *last-iterate convergence* guarantee of various games including IIEFGs (Lin et al., 2020; Wei et al., 2021a;a; Lee et al., 2021; Cai et al., 2022; Abe et al., 2023; Feng et al., 2023; Cen et al., 2023; Liu et al., 2023). Specifically, Lee et al. (2021) and Liu et al. (2023) establish algorithms for learning IIEFGs with a last-iterate convergence rate of $\widetilde{\mathcal{O}}(1/k)$. However, the algorithms proposed by Lee et al. (2021) and Liu et al. (2023) rely on full-information feedback when learning IIEFGs, and therefore cannot be directly applied in practical scenarios where only bandit feedback is available. The above considerations naturally motivate the following question:

*Can we achieve last-iterate convergence for learning IIEFGs with bandit feedback?*

In fact, this same question has also been posed by Fiegel et al. (2023). In this work, we answer this question affirmatively. The main contributions of our work are summarized as follows:

- We introduce the first algorithm that learns the approximate NE of IIEFGs with provable last-iterate convergence in the bandit feedback setting. Unlike previous approaches (Lee et al., 2021; Liu et al., 2023) that use a dilated negentropy regularizer to achieve last-iterate convergence for IIEFGs under full-information feedback, our algorithm employs a negentropy regularizer weighted by a "virtual transition" over the information set-action space. This design prevents our algorithm from encountering the issues of coupled information set-action pairs—a problem that arises with the dilated negentropy regularizer and would otherwise result in an excessively large stability term when applying the entropy regularization technique over the objective function. On the other hand, from the computational perspective, we show that our algorithm still admits a closed-form solution for policy updates over the full policy space, similar to the dilated negentropy regularizer. This facilitates efficient approximate policy updates, in contrast to using the vanilla negentropy regularizer (see Section 4.1 for details). Furthermore, our algorithm does not require any communication or coordination between the two players and is model-free, without requiring the knowledge of the underlying state transition probabilities.

- To achieve the last-iterate convergence rate with a sharp dependence on $X$ (and $Y$), it is essential to establish an efficient bound for the stability term of OMD with the virtual transition-weighted negentropy regularizer. To this end, we devise a virtual transition over the information set-action space that maximizes the minimum "visitation probability" across all information sets (see Section 4.2 for more details). With this virtual transition, we ultimately prove that our algorithm achieves a last-iterate convergence rate for learning IIEFGs in the bandit feedback setting of $\widetilde{\mathcal{O}}((X + Y)[(XA + YB)^{1/2} + (X + Y)^{1/4}H]k^{-1/8})$ with high probability for each episode $k$. When only obtaining an expected NE gap is of interest, we also show that our algorithm can generate a policy profile that converges to the NE with a rate of $\widetilde{\mathcal{O}}((X + Y)[(X^2A + Y^2B)^{1/2} + (X + Y)^{1/4}H]k^{-1/6})$ (see Section 5.1 for more details).

- Additionally, we conduct empirical evaluations on a variety of IIEFG instances, which demonstrate the advantages of our proposed algorithm over baseline methods (see Appendix I for details).

## 2. Related Works

### 2.1. Partially Observable Markov Games (POMGs)

With perfect information, learning Markov games (MGs) can be traced back to the seminal work of Littman & Szepesvári (1996) and has since garnered extensive research attention (Littman, 2001; Greenwald & Hall, 2003; Hu & Wellman, 2003; Hansen et al., 2013; Sidford et al., 2018; Pérolat et al., 2015; Fan et al., 2020; Jia et al., 2019; Cui & Yang, 2021; Zhang et al., 2021; Bai & Jin, 2020; Liu et al., 2021; Zhou et al., 2021; Song et al., 2022; Li et al., 2022; Xiong et al., 2022; Wang et al., 2023; Cui et al., 2023). In scenarios where only imperfect information is available yet the complete knowledge of the game (state transitions and rewards) is known, existing research can be categorized into three primary streams. The first stream leverages sequence-form policies to recast the problem as a linear program (Koller & Megiddo, 1992; Von Stengel, 1996; Koller et al., 1996). The second stream translates the problem into a minimax optimization problem and explores first-order algorithms, as exemplified in Hoda et al. (2010); Kroer et al. (2015; 2018); Munos et al. (2020); Lee et al. (2021); Liu et al. (2022). Lastly, the third stream addresses the problem through CFR, minimizing counterfactual regrets locally within each information set (Zinkevich et al., 2007; Lanctot et al., 2009; Johanson et al., 2012; Tammelin, 2014; Schmid et al., 2019; Burch et al., 2019; Liu et al., 2022).

In the realm where the knowledge of the game is unknown, existing research focuses on integrating OMD and FTRL frameworks with importance-weighted loss estimators (Farina et al., 2021; Kozuno et al., 2021; Bai et al., 2022; Fiegel et al., 2023). Remarkably, Bai et al. (2022) achieve the sample complexity of $\widetilde{\mathcal{O}}\left(H^3(XA + YB)/\varepsilon^2\right)$ for learn-

ing an $\varepsilon$-approximate NE by employing a "balanced" dilated KL-divergence as the distance metric. Building upon this concept, Fiegel et al. (2023) utilize "balanced transitions" and attain a (nearly) optimal sample complexity of $\widetilde{\mathcal{O}}\left(H(XA + YB)/\varepsilon^2\right)$. However, we note that all the existing algorithms studying POMGs with bandit feedback only have *average-iterate convergence* guarantees, while we aim to establish the algorithms with *last-iterate convergence* guarantees.

## 2.2. Last-iterate Convergence Learning in Games

With full-information feedback, learning in games with last-iterate convergence guarantee has been investigated in strongly monotone games (Mokhtari et al., 2020; Jordan et al., 2024), monotone games (Golowich et al., 2020; Cai et al., 2022; Gorbunov et al., 2022; Cai & Zheng, 2023), Markov games (Cen et al., 2021; Zeng et al., 2022; Cen et al., 2023), and IIEFGs (Lee et al., 2021; Liu et al., 2023; Bernasconi et al., 2024). Besides, there are some works studying achieving last-iterate convergence in games with noisy feedback. For instance, Abe et al. (2023; 2024; 2025) establish algorithms for solving two-player zero-sum matrix games or multi-player monotone games with noisy gradient feedback, where the noisy feedback for all the actions is observable.

Recently, motivated by the fact that it might be restrictive to require full knowledge of the (noisy) gradient as in the full-information feedback setting, a growing body of works has studied learning in games with last-iterate convergence guarantee in the bandit feedback setting, including strongly monotone games (Bravo et al., 2018; Hsieh et al., 2019; Lin et al., 2021; Drusvyatskiy et al., 2022; Huang & Hu, 2023) and matrix games (Cai et al., 2023). Though the last-iterate convergence guarantees have also been established for Markov games (Wei et al., 2021b; Chen et al., 2022; 2023; Cai et al., 2023), existing algorithms are not fully decoupled, with the exception of the algorithms of Chen et al. (2023) and Cai et al. (2023). In particular, the algorithm of Wei et al. (2021b) needs coordinated updates and prior knowledge of the game, and the algorithm of Chen et al. (2022) requires the players to inform the opponent about the entropy of their own policies. Moreover, we note that all existing works study fully observable Markov games, while this work aims to establish uncoupled algorithms for learning IIEFGs in the formulation of partially observable Markov games, where only partial information of the underlying states is revealed to the players.

## 3. Preliminaries

For ease of exposition, we consider IIEFGs in the formulation of POMGs (Kozuno et al., 2021; Bai et al., 2022) and introduce the preliminaries in this section.

**Partially Observable Markov Games**  We study episodic, finite-horizon, two-player zero-sum POMGs, denoted by $\text{POMG}(H, \mathcal{S}, \mathcal{X}, \mathcal{Y}, \mathcal{A}, \mathcal{B}, \mathbb{P}, r)$, in which

- $H$ is the horizon length;

- $\mathcal{S} = \bigcup_{h \in [H]} \mathcal{S}_h$ is the finite state space, where $\mathcal{S}_h$ is the state space at step $h$ and $\mathcal{S}_h \bigcap \mathcal{S}_{h'} = \emptyset$ for any $h \neq h'$. $S = \sum_{h=1}^{H} S_h$ is the size of $\mathcal{S}$ and $|\mathcal{S}_h| = S_h$ for all $h$;

- $\mathcal{X} = \bigcup_{h \in [H]} \mathcal{X}_h$ is the finite space of information sets (short for *infosets* in the following) for the max-player, where $\mathcal{X}_h = \{x(s) : s \in \mathcal{S}_h\}$ is the set of the infosets at step $h$ and $x : \mathcal{S} \rightarrow \mathcal{X}$ is the emission function. $X = \sum_{h=1}^{H} X_h$ is the size of $\mathcal{X}$ with $|\mathcal{X}_h| = X_h$. The finite space of infosets $\mathcal{Y} = \bigcup_{h \in [H]} \mathcal{Y}_h$ for the min-player and its size are defined analogously;

- $\mathcal{A}$ with $|\mathcal{A}| = A$ and $\mathcal{B}$ with $|\mathcal{B}| = B$ are the finite action spaces for the max-player and min-player, respectively;

- $\mathbb{P} = \{p_0(\cdot) \in \Delta_{\mathcal{S}_1}\} \bigcup \{p_h(\cdot|s_h, a_h, b_h) \in \Delta_{\mathcal{S}_{h+1}}\}_{(s_h, a_h, b_h) \in \mathcal{S}_h \times \mathcal{A} \times \mathcal{B}, h \in [H-1]}$ are the state transition probabilities, where $p_0(\cdot)$ is the probability distribution of initial states, $p_h(s_{h+1}|s_h, a_h, b_h)$ is the probability of transitioning to the next state $s_{h+1}$ conditioned on $(s_h, a_h, b_h)$ at step $h$, and $\Delta_{\mathcal{S}_h}$ denotes the probability simplex over $\mathcal{S}_h$;

- $r = \{r_h(s_h, a_h, b_h) \in [0, 1]\}_{(s_h, a_h, b_h) \in \mathcal{S}_h \times \mathcal{A} \times \mathcal{B}, h \in [H]}$ are the (random) reward functions with $\bar{r}_h(s_h, a_h, b_h)$ as mean for each $r_h(s_h, a_h, b_h)$.

**Learning Protocol**  We define the max-player's (stochastic) policy as $\mu = \{\mu_h\}_{h \in [H]}$, where $\mu_h : \mathcal{X}_h \rightarrow \Delta_{\mathcal{A}}$ denotes the max-player's policy at step $h$. The set of all such policies for the max-player is denoted by $\Pi_{\max}$. Analogously, the min-player's (stochastic) policy is denoted by $\nu = \{\nu_h\}_{h \in [H]}$, with $\nu_h : \mathcal{Y}_h \rightarrow \Delta_{\mathcal{B}}$ being the min-player's policy at step $h$, and the set of all min-player's policies is denoted by $\Pi_{\min}$. The game proceeds in a finite number of episodes. At the commencement of episode $k$, the max-player selects a policy $\mu^k \in \Pi_{\max}$, while the min-player chooses $\nu^k \in \Pi_{\min}$. Meanwhile, an initial state $s_1^k$ is sampled from $p_0(\cdot)$ by the environment. During each step $h$ within episode $k$, the max-player and min-player observe their respective infosets $x_h^k := x(s_h^k)$ and $y_h^k := y(s_h^k)$, but they do not observe the underlying state $s_h^k$. Given $x_h^k$, the max-player takes an action $a_h^k \sim \mu_h^k(\cdot|x_h^k)$, while the min-player concurrently takes an action $b_h^k \sim \nu_h^k(\cdot|y_h^k)$. Upon taking these actions, the max-player and min-player receive rewards $r_h^k := r_h(s_h^k, a_h^k, b_h^k)$ and $-r_h^k$, respectively. Subsequently, the game transitions to the next state $s_{h+1}^k \sim p_h(\cdot|s_h^k, a_h^k, b_h^k)$. The $k$-th episode will terminate after actions $a_H^k$ and $b_H^k$ are taken conditioned on $x_H^k$ and $y_H^k$ and rewards $r_H^k$ and $-r_H^k$ are observed by the max-player and min-player, respectively.

**Perfect Recall and Tree Structure** Following prior works (Kozuno et al., 2021; Bai et al., 2022; Fiegel et al., 2023), we assume that the POMGs adhere to the *tree structure* and the *perfect recall* condition (Kuhn, 1953). The tree structure signifies that for any step $h = 2, \ldots, H$ and state $s_h \in \mathcal{S}_h$, there exists a *unique path* $(s_1, a_1, b_1, \ldots, s_{h-1}, a_{h-1}, b_{h-1})$ culminating in $s_h$. The perfect recall condition, meanwhile, is fulfilled for both players, implying that for any $h = 2, \ldots, H$ and any infoset $x_h \in \mathcal{X}_h$, there exists a *unique* history $(x_1, a_1, \ldots, x_{h-1}, a_{h-1})$ leading to $x_h$ (analogously for the min-player). Furthermore, we denote by $C_{h'}(x_h, a_h) \subset \mathcal{X}_{h'}$ the set of descendants of the infoset-action pair $(x_h, a_h)$ at step $h' \geq h$. Also, we define $C_{h'}(x_h) := \bigcup_{a_h \in \mathcal{A}} C_{h'}(x_h, a_h)$ as the union of descendants across all actions at $x_h$. For convenience, let $C(x_h, a_h) := C_{h+1}(x_h, a_h)$ signify the immediate descendants at the subsequent step.

**Sequence-form Representations** For any pair of product policies $(\mu, \nu)$, the tree structure and the perfect recall condition facilitate a *sequence-form representation* of the reaching probability for the state-action tuple $(s_h, a_h, b_h)$: $\mathbb{P}^{\mu,\nu}(s_h, a_h, b_h) = p_{1:h}(s_h)\mu_{1:h}(x(s_h), a_h)\nu_{1:h}(y(s_h), b_h)$, where $p_{1:h}(s_h) = p_0(s_1)\prod_{h'=1}^{h-1} p_{h'}(s_{h'+1}|s_{h'}, a_{h'}, b_{h'})$ denotes the sequence-form transition probability, and $\mu_{1:h}(x_h, a_h) := \prod_{h'=1}^{h} \mu_{h'}(a_{h'}|x_{h'})$ and $\nu_{1:h}(y_h, b_h) := \prod_{h'=1}^{h} \nu_{h'}(b_{h'}|y_{h'})$ represent the sequence-form policies of the max-player and min-player, respectively. Under the sequence-form representation, we adopt a slight abuse of notation for $\mu$ and $\nu$ by interpreting them as $\mu = \{\mu_{1:h}\}_{h \in [H]}$ and $\nu = \{\nu_{1:h}\}_{h \in [H]}$.[1] Furthermore, it is clear that $\Pi_{\max}$ constitutes a convex compact subspace of $\mathbb{R}^{XA}$ that adheres to the constraints $\mu_{1:h}(x_h, a_h) \geq 0$ and $\sum_{a_h \in \mathcal{A}} \mu_{1:h}(x_h, a_h) = \mu_{1:h-1}(x_{h-1}, a_{h-1})$, where $(x_{h-1}, a_{h-1})$ is such that $x_h \in C(x_{h-1}, a_{h-1})$ (with the convention that $\mu_{1:0}(x_0, a_0) = 1$ as a base case).

**Learning Objective** In this work, we consider the learning objective of finding an approximate NE of the POMGs. Specifically, for any $\varepsilon \geq 0$, an $\varepsilon$-approximate NE is a pair of product policy $(\mu, \nu)$ satisfying $\mathrm{NEGap}(\mu, \nu) \leq \varepsilon$, where

$$\mathrm{NEGap}(\mu, \nu) := \sup_{\mu^\dagger \in \Pi_{\max}, \nu^\dagger \in \Pi_{\min}} V^{\mu^\dagger, \nu} - V^{\mu, \nu^\dagger}, \quad (1)$$

and $V^{\mu,\nu} = \mathbb{E}_{\mu,\nu}\left[\sum_{h=1}^{H} r_h(s_h, a_h, b_h)\right]$ is the value function of $(\mu, \nu)$ with the expectation taken over the randomness of the product policy pair $(\mu, \nu)$ and the environment (*i.e.*, $\mathbb{P}$ and $r$). It is known that using regret to NE conversion, an approximate NE can be obtained by averaging all

---

[1] The set of sequence-form policies is defined in a top-down manner and is equivalent to the "treeplex" space of policies defined in a bottom-up manner (see, *e.g.*, Lee et al. (2021)).

the policies $\{\mu\}_{k=1}^{K}$ of the max-player generated by an algorithm with sublinear regret (similarly for the min-player) to obtain the average policy pair $(\bar{\mu}, \bar{\nu})$ (see, *e.g.*, Theorem 1 of Kozuno et al. (2021)). This is the so-called *average-iterate convergence* of learning NE. By contrast, as explained in Section 1, in this work, we are interested in finding the $\varepsilon$-NE with the (finite-time) *last-iterate convergence* guarantee; that is, the algorithm is required to generate an approximate NE policy profile $(\mu^k, \nu^k)$ such that $\mathrm{NEGap}(\mu^k, \nu^k) \leq \varepsilon_k$ for each (finite-time) episode $k$.

**Information Available to the Players** In this work, learning POMGs in the bandit feedback setting is considered. Specifically, in each episode $k$, the max-player only observes her experienced trajectory $(x_1^k, a_1^k, r_1^k, \ldots, x_H^k, a_H^k, r_H^k)$ of infosets, actions, and rewards, but not the underlying states or the opponent's infosets and actions (similarly for the min-player). Additionally, the max-player has no knowledge of the min-player's policies and cannot receive any information from the min-player, and vice versa. Besides, there is no shared randomness between both players; that is, the algorithms of both players need to be fully uncoupled from each other.

**Additional Notations** We slightly abuse the notation to view $x_h$ as the set $\{s \in \mathcal{S}_h : x(s) = x_h\}$, when writing $s \in x_h$. Given sequence-form representations, for any $\mu \in \Pi_{\max}$ and a sequence of functions $f = (f_h)_{h \in [H]}$ with $f_h : \mathcal{X}_h \times \mathcal{A} \to \mathbb{R}$, we define $\langle \mu, f \rangle := \sum_{h \in [H], (x_h, a_h) \in \mathcal{X}_h \times \mathcal{A}} \mu_{1:h}(x_h, a_h)f_h(x_h, a_h)$. We denote by $\mathcal{F}^k$ the $\sigma$-algebra generated by the random variables $\{(s_h^t, a_h^t, b_h^t, r_h^t)\}_{h \in [H], t \in [k]}$. For brevity, we abbreviate the conditional expectation $\mathbb{E}[\cdot \mid \mathcal{F}^k]$ as $\mathbb{E}^k[\cdot]$. Throughout this paper, the notation $\widetilde{\mathcal{O}}(\cdot)$ suppresses all logarithmic factors.

## 4. Algorithm

This section presents the proposed algorithm, detailed in Algorithm 1. In Section 4.1, we introduce the algorithmic framework as well as the virtual transition-weighted negentropy regularizer. In Section 4.2, we present the algorithmic design to compute an effective virtual transition to incorporate into the regularizer.

### 4.1. From Sequence-form Policies to Probability Measures over Infoset-Action Space

With sequence-form policies, we first reformulate the IIEFG into the following bilinear game:

$$f(\mu, \nu) = \mu^\top \boldsymbol{G} \nu, \quad (3)$$

where $\boldsymbol{G} \in \mathbb{R}^{XA \times YB}$ is the loss matrix with $\boldsymbol{G}((x_h, a_h), (y_h, b_h)) = \sum_{s_h \in x_h \cap y_h} p_{1:h}(s_h)(1 - r_h(s_h, a_h, b_h))$. In

---

**Algorithm 1** OMD with Virtual Transition-Weighted Negentropy Regularization (max-player)

---

1: **Input:** $\eta_k = k^{-\alpha_\eta}, \gamma_k = k^{-\alpha_\gamma}, \varepsilon_k = k^{-\alpha_\varepsilon}$.
2: **Initialize:** $\mu_1(a_h|x_h) = \frac{1}{A}, \forall (x_h, a_h) \in \mathcal{X}_h \times \mathcal{A}, \forall h \in [H]$. Set virtual transition $p^x$ computed by Algorithm 2.
3: **for** $k = 1, \cdots,$ **do**
4:     **for** $h = 1, \cdots, H$ **do**
5:         Observes $x_h^k$, executes $a_h^k \sim \mu_h^k(\cdot|x_h^k)$ and receives $r_h^k$.
6:         For all $(x_h, a_h) \in \mathcal{X}_h \times \mathcal{A}$, sets entropy regularized loss estimator as in Eq. (7).
7:     **end for**
8:     Update policy

$$\mu^{k+1} = \arg\min_{\mu \in \Pi_{\max}^{k+1}} \eta_k \langle \mu, \widehat{\ell}^k \rangle + D_\psi(\mu, \mu^k), \quad (2)$$

    where $\Pi_{\max}^{k+1} = \{\mu \in \Pi_{\max} : \mu(a_h|x_h) \geqslant \frac{1}{A(k+1)}, \forall (x_h, a_h) \in \mathcal{X}_h \times \mathcal{A}, \forall h \in [H]\}$.
9: **end for**

---

**Algorithm 2** Computing Virtual Transition $p^x$ (max-player)

---

1: **Input:** Game tree structure of $\mathcal{X} \times \mathcal{A}$.
2: **Initialize:** Sequence-form representation of virtual transition $q \in \mathbb{R}^X$. Array of maximized number of descendant infosets $c \in \mathbb{R}^X, d \in \mathbb{R}^{XA}$. For all $x_H$ in $\mathcal{X}_H$, set $c(x_H) = 1$.
3: **for** $h = H - 1$ to $1$ **do**
4:     **for** $x_h$ in $\mathcal{X}_h$ **do**
5:         **for** $a_h$ in $\mathcal{A}$ **do**
6:             Compute $d(x_h, a_h) = \sum_{x_{h+1} \in C(x_h, a_h)} c(x_{h+1})$.
7:         **end for**
8:         Compute $c(x_h) = \max_{a \in \mathcal{A}} d(x_h, a)$.
9:     **end for**
10: **end for**
11: **for** $x_1$ in $\mathcal{X}_1$ **do**
12:     Compute $q_{1:1}(x_1) = \frac{c(x_1)}{\sum_{x_1 \in \mathcal{X}_1} c(x_1)}$.
13: **end for**
14: **for** $h = 1$ to $H - 1$ **do**
15:     **for** $x_h, a_h$ in $\mathcal{X}_h \times \mathcal{A}$ **do**
16:         **for** $x_{h+1}$ in $C(x_h, a_h)$ **do**
17:             Compute $q_{1:h+1}(x_{h+1}) = q_{1:h}(x_h) \cdot \frac{c(x_{h+1})}{\sum_{x'_{h+1} \in C(x_h, a_h)} c(x'_{h+1})}$.
18:         **end for**
19:     **end for**
20: **end for**
21: **return** $q$.

---

this manner, the learning objective is equivalent to finding $(\mu, \nu)$ such that $\text{NEGap}(\mu, \nu) =$

$\sup_{\mu^\dagger \in \Pi_{\max}, \nu^\dagger \in \Pi_{\min}} f(\mu, \nu^\dagger) - f(\mu^\dagger, \nu) \leq \varepsilon$. At a high level, we apply the entropy regularizing technique to perturb the bilinear form of the game in Eq. (3) into a strongly convex-strongly concave structure, ensuring convergence to the NE of the perturbed game (and thus the NE of the original game). This approach builds upon previous research that has explored last-iterate convergence learning in Markov games with full-information feedback (Cen et al., 2021; Chen et al., 2022; Cen et al., 2023) and with bandit feedback (Cai et al., 2023), and IIEFGs with full-information feedback (Liu et al., 2023). In detail, we consider the following perturbed game as a surrogate:

$$f_k(\mu, \nu) = \mu^\top \boldsymbol{G} \nu + \varepsilon_k \psi(\mu) - \varepsilon_k \psi(\nu), \quad (4)$$

where $\psi$ is some strongly convex regularizer used in OMD and $\varepsilon_k > 0$ serves as the knob to control the strength of the entropy regularization in episode $k$. By gradually decreasing $\varepsilon_k$ to be moderately small, the approximate NE of the perturbed game in Eq. (4) will also serve as an approximate NE of the original game in Eq. (3).

The crucial aspect lies in selecting an appropriate regularizer $\psi$. A natural approach is leveraging the dilated negentropy (Kroer et al., 2015; Kozuno et al., 2021):

$$\psi(\mu) = \sum_{h, x_h, a_h} \mu_{1:h}(x_h, a_h) \log \left( \frac{\mu_{1:h}(x_h, a_h)}{\mu_{1:h}(x_h)} \right), \quad (5)$$

which has been widely used in existing literature studying IIEFGs (Kozuno et al., 2021; Lee et al., 2021; Liu et al., 2023; Bernasconi et al., 2024). In particular, the dilated negentropy has been used to achieve the last-iterate convergence in IIEFGs with full-information feedback (Lee et al., 2021; Liu et al., 2023; Bernasconi et al., 2024). However, the defect of dilated negentropy is that infoset-action pairs on different steps are actually coupled with each other (recall $\mu_{1:h}(x_h) = \sum_{a_h \in \mathcal{A}} \mu_{1:h}(x_h, a_h)$ in Eq. (5)). In IIEFGs with bandit feedback, leveraging OMD using dilated negentropy together with the entropy regularization technique will deduce a stability term scaling with $\mathbb{E}_{z^k}[\exp(-\sum_{h, x_h, a_h} z_{1:h}^k(x_h, a_h) \log \mu^k(a_h|x_h))]$, where $z^k(x_h, \cdot) \sim \text{Cat}(\mu_h^k(\cdot|x_h))$ is a random vector independently sampled from the categorical distribution parameterized by $\mu_h^k(\cdot|x_h)$ and $z_{1:h}^k(x_h, a_h) = \prod_{i=1}^h z^k(x_i, a_i)$ is the sequence-form representation of $z^k$. As $\log \mu^k(a_h|x_h)$ contributed by entropy regularization might be potentially very negative, this renders this upper bound of the stability term vacuous and is not sufficient to obtain a meaningful last-iterate convergence rate.[2]

---

[2]It should be noted that this issue cannot be resolved by merely restricting the feasible set to a subset of $\Pi_{\max}$. In the case of OMD with dilated negentropy, even if the feasible set is constrained as $\Pi_{\max}^k = \{\mu \in \Pi_{\max} : \mu(a_h|x_h) \geqslant \frac{1}{Ak}\}$—the approach used in our Algorithm 1—the stability term can still reach magnitudes as large as $\exp(\mathcal{O}(X))$, which is prohibitively large.

To cope with this issue, we instead consider using the negentropy regularizer weighted by a kind of "virtual transition" $p^x$ over the infoset-action space $\mathcal{X} \times \mathcal{A}$:

$$\psi_{p^x}(\mu) = \sum_{h, x_h, a_h} [p^x \mu](x_h, a_h) \log [p^x \mu](x_h, a_h), \quad (6)$$

where $[p^x \mu](x_h, a_h) = p^x_{1:h}(x_h) \mu_{1:h}(x_h, a_h)$, $p^x_h(\cdot | x_h, a_h) \in \Delta_{C(x_h, a_h)}$ is a transition probability over $\mathcal{X}_h \times \mathcal{A} \times \mathcal{X}_{h+1}$, and $p^x_{1:h}(x_h) = p^x_0(x_1) \prod_{h'=1}^{h-1} p^x_{h'}(x_{h'+1} | x_{h'}, a_{h'})$ is its sequence-form representation. Note that $p^x_h(x_{h+1} | x(s_h), a_h)$ is not necessarily the true transition probability $\mathbb{P}^{\mu^k, \nu^k}(x_{h+1} | x(s_h), a_h) = \sum_{s_{h+1} \in x_{h+1}, b_h \in \mathcal{B}} p(s_{h+1} | s_h, a_h, b_h) \nu^k(b_h | y(s_h))$ experienced by the max-player in episode $k$. Also, notice that the constructed virtual transition $p^x$ is well-defined by the perfect recall condition and $p^x \mu$ is a probability measure over the infoset-action space $\mathcal{X}_h \times \mathcal{A}$ at step $h$. Therefore, by incorporating the virtual transition $p^x$, we actually regularize the probability measures over $\mathcal{X} \times \mathcal{A}$ instead of directly regularizing the sequence-form policy $\mu$, bypassing the issue arising when dealing with the coupled (and potentially very negative) loss estimates of infoset-action pairs of dilated negentropy. For notational convenience, we drop the dependence in the subscript of $\psi_{p^x}(\cdot)$ on $p^x$, when the context is clear for brevity.

With regularizer $\psi$ specified, the derivative of $f_k(\mu, \nu)$ w.r.t. $\mu(x_h, a_h)$ is $\frac{\partial f_k(\mu, \nu)}{\partial \mu_{1:h}(x_h, a_h)} = [\boldsymbol{G}\nu](x_h, a_h) + \varepsilon_k \cdot p^x_{1:h}(x_n)(\log [p^x \mu](x_h, a_h) + 1)$. Since $[p^x \mu] \in \prod_{h=1}^{H} \Delta_{\mathcal{X}_h \times \mathcal{A}}$ for any $\mu$, the constant $1$ in the above display does not affect the optimization of OMD. Besides, with bandit feedback, an (optimistically biased) loss estimate $\frac{\mathbb{I}^k_h \{x_h, a_h\}}{\mu^k_{1:h}(x_h, a_h) + \gamma_k}(1 - r^k_h)$ of $[\boldsymbol{G}\nu](x_h, a_h)$ in episode $k$ is constructed (Kozuno et al., 2021), where $\gamma_k > 0$ is the implicit exploration parameter (Neu, 2015) and $\mathbb{I}^k_h \{x_h, a_h\} := \mathbb{I}\{(x_h, a_h) = (x^k_h, a^k_h)\}$. This specifies the final entropy regularized loss estimator as follows (Line 1):

$$\widehat{\ell}^k_h(x_h, a_h) = \frac{\mathbb{I}^k_h \{x_h, a_h\}}{\mu^k_{1:h}(x_h, a_h) + \gamma_k}(1 - r^k_h) \\ + \varepsilon_k \cdot p^x_{1:h}(x_h) \log [p^x \mu^k](x_h, a_h). \quad (7)$$

With the constructed loss estimator, Algorithm 1 then uses OMD to update policy. Since now the entropy regularized loss estimator is considered, the variance of the loss estimator scales with $|\log [p^x \mu](x_h, a_h)|$ and will be prohibitively large if running OMD on the entire $\Pi_{\max}$, eventually leading to an unbounded stability term of OMD. Hence we constrain the feasible set of the OMD as a subset $\Pi^{k+1}_{\max}$ of $\Pi_{\max}$, where each $\mu \in \Pi^{k+1}_{\max}$ satisfies $\mu(a_h | x_h)$ is lower bounded for all $(x_h, a_h) \in \mathcal{X}_h \times \mathcal{A}$ and $h \in [H]$ (Line 1).

**Computation** Since the update of OMD is now constrained onto a subset $\Pi^k_{\max}$ of $\Pi_{\max}$, the computation of Eq. (2) generally does not have a closed-form solution. However, we note that the approximate update of our Algorithm 1 still admits an efficient closed-form solution. Specifically, notice that the operation of constraining the update of OMD onto the constrained set $\Pi^k_{\max}$ is only for the aim of preventing the entropy regularized loss from being prohibitively large. In practice, we can still compute $\mu^k$ over the whole $\Pi_{\max}$, but clip the regularized loss when it becomes undesirably large. Importantly, in Appendix A, we prove that updating $\mu^k$ over the whole $\Pi_{\max}$ has a closed-form solution. As shown in the experiments, clipping the loss estimator and then operating OMD in the whole $\Pi_{\max}$ using this closed-form update suffices to obtain an appealing performance (please see Appendix I for the experiment details).

### 4.2. Virtual Transition with Maximized Minimum Visitation Probability

As elaborated in Section 4.1, our Algorithm 1 leverages a virtual transition-weighted negentropy to regularize the loss estimator and induce the Bregman divergence $D_\psi(\cdot, \cdot)$ used in OMD. The upside of employing such virtual transition $p^x$ lies in that it implicitly helps to operate the update of OMD in the space of probability measures over infoset-action pairs instead of the sequence-form policies, to avoid the coupling between different infoset-action pairs. However, this is still not sufficient to obtain a well-controlled stability term if additional care is not taken. Specifically, upon applying the virtual transition to weight the negentropy, the stability term associated with OMD at each infoset $x_h$ will be enlarged by (approximately) a multiplicative factor of $1/p^x_{1:h}(x_h)$. This enlargement arises intuitively from the fact that, $D_\psi(\cdot, \cdot)$ induced by $\psi$ at each $x_h$ undergoes a downscaling, proportional to $p^x_{1:h}(x_h)$, thereby resulting in a relative increase in the stability term. Therefore, to ensure that the stability term is well-controlled, we design the following $p^x$ which maximizes the minimum "visitation probability" of all $x_h$ in its sequence-form representation:

$$p^x = \arg\max_{q \in \mathbb{P}^x} \min_{x_h \in \mathcal{X}_h, h \in [H]} q_{1:h}(x_h), \quad (8)$$

where $\mathbb{P}^x$ denotes the set of all the valid virtual transitions over infoset-action space. The solution to Eq. (8) can be computed efficiently via backward dynamic programming, as shown in Algorithm 2.

Note that similar ideas leveraging negentropy weighted by the transition over infoset-action space have also been exploited by Bai et al. (2022); Fiegel et al. (2023). However, we would like to underscore that the design of our virtual transition $p^x$ over infoset-action space is different from those of Bai et al. (2022); Fiegel et al. (2023). In detail, our virtual transition $p^\star(x_{h+1} | x_h, a_h)$ is defined based on

some fixed action $a_{h+1} \in \mathcal{A}$ that maximizes the number of reachable infosets $|C_H(x_{h+1}, a_{h+1})| \in \mathcal{X}_H$. This approach contrasts with the "balanced transitions" introduced by Bai et al. (2022) and Fiegel et al. (2023), which either consider all reachable infosets in a specific layer $\mathcal{X}_{h'}$ for some $h' \geq h + 1$, the entire sub-tree, or compute transitions as the sum of reachable infosets across all possible actions $a_{h+1} \in \mathcal{A}$ at infoset $x_{h+1}$. Further, we aim to establish the last-iterate convergence of IIEFGs while they can only guarantee the average-iterate convergence, necessitating different theoretical analysis.

**On the Requirement of Knowing Game Tree Structure**
The construction of our virtual transition by Algorithm 2 requires the game tree structure to be known a priori, which is also required by some algorithms learning IIEFGs with average-iterate convergence (Bai et al., 2022; Fiegel et al., 2023). While there are algorithms with average-iterate convergence that do not need prior knowledge of the tree structure (*e.g.*, Kozuno et al. (2021)), we note the game tree structure can be extracted from one traversal on the game tree in $O(XA)$ time. Therefore, this requirement is mild on game instances with moderately large $X$ and $A$ (Bai et al., 2022). Whereas, we also remark that there exist some game instances with exponentially large $X$ (*e.g.*, no-limit Texas hold'em (Johanson, 2013)), making one traversal on the game tree impractical. In such cases, the polynomial dependence on $X$ of the convergence rate lower bound (see Section 5.2) indicates that it is also statistically intractable to learn such game instances if no function approximation assumptions are further imposed. Besides, in cases of unknown game tree structure, we show that using vanilla negentropy is also able to achieve the last-iterate convergence in IIEFGs (see Remark 5.4 and Appendix F for details). This approach no longer requires knowledge of the game tree structure. Nevertheless, the downside of using OMD with vanilla negentropy is that it does not admit a closed-form update (Hoda et al., 2010), even though the update of OMD is performed on the whole $\Pi_{\max}$, since it does not adapt to the game tree structure.

**Experiments** As aforementioned, our Algorithm 1 admits an efficient approximate policy update. We conduct empirical evaluations on various IIEFG game instances, including Lewis Signaling, Kuhn Poker (Kuhn, 1950), Leduc Poker (Southey et al., 2012), and Liars Dice. The empirical evaluations show that when Algorithm 1 is equipped with the virtual transition computed by Algorithm 2, it can perform relatively well across all game instances. Though there might be some baseline that performs similarly to our algorithm on some game instances, this baseline algorithm might not be able to converge fast on other game instances, as the last-iterate convergences of all the baseline algorithms are not theoretically guaranteed. We defer the detailed ex-

perimental results to Appendix I due to space limit.

## 5. Analysis

In Section 5.1, we first present the upper bound of the last-iterate convergence rate of our Algorithm 1. Then in Section 5.2, we provide the lower bound for learning IIEFGs with bandit feedback and last-iterate convergence guarantee.

### 5.1. Upper Bound of Last-iterate Convergence

**Theorem 5.1.** *If Algorithm 1 is adopted by both players, by setting $\alpha_\eta = 5/8$, $\alpha_\gamma = 3/8$ and $\alpha_\varepsilon = 1/8$, for any $k \geq 1$, with probability at least $1 - \widetilde{\mathcal{O}}(\delta)$, it holds that*

$$
\begin{aligned}
&\mathrm{NEGap}(\mu^k, \nu^k) \\
=& \widetilde{\mathcal{O}}\left( \left[ (XA + YB)^{\frac{1}{2}} k^{-\frac{1}{8}} + (XA + YB)^{\frac{1}{2}} H k^{-\frac{3}{8}} \right.\right.\\
&\left.\left. + \left( X^2 A + Y^2 B \right)^{\frac{1}{2}} k^{-\frac{1}{4}} + (X + Y)^{\frac{1}{4}} H k^{-\frac{1}{8}} \right] (X + Y) \right).
\end{aligned}
$$

**Remark 5.2.** *When $k \geq \max\{H^4, \left(X^2 A + Y^2 B\right)^4 / (XA + YB)^4, (XA + YB)^{8/7} / (X+Y)^{10/7}\}$, we have $\mathrm{NEGap}(\mu^k, \nu^k) = \widetilde{\mathcal{O}}((X + Y)[(XA + YB)^{1/2} + (X + Y)^{1/4} H]k^{-1/8})$. Besides, when only obtaining an expected last-iterate convergence rate is desired, our Algorithm 1 has an improved last-iterate convergence rate of $\widetilde{\mathcal{O}}((X + Y)[(X^2 A + Y^2 B)^{1/2} + (X + Y)^{1/4} H]k^{-1/6})$ in expectation (see Appendix E for details).*

**Remark 5.3.** *Though the last-iterate convergence rate of our Algorithm 1 is inferior to the $\widetilde{\mathcal{O}}(1/k)$ convergence rate by Lee et al. (2021); Liu et al. (2023), we note that both their algorithms can only work in the full-information setting. Further, we remark that the algorithm of Lee et al. (2021) needs the assumption that the NE of the IIEFG considered is unique. Though such an assumption is not required by Liu et al. (2023), the algorithm of Liu et al. (2023) requires both players to be controlled by a central controller, and thus their algorithm is not uncoupled. In contrast, our algorithm can work in the bandit feedback setting, is fully uncoupled between the two players, and can still guarantee a regret of order $\widetilde{\mathcal{O}}(k^{7/8})$ even when the opponent of the max-player is an adversary. Moreover, Section 5.2 shows that the lower bound of the convergence rate for learning IIEFGs with bandit feedback, last-iterate convergence guarantee, and uncoupled algorithms will be of order $\Omega(k^{-1/2})$ (for large enough $k$).*

**Remark 5.4.** *In Appendix F, we demonstrate that employing vanilla negentropy $\psi(\mu) = \sum_{h, x_h, a_h} \mu_{1:h}(x_h, a_h) \log \mu_{1:h}(x_h, a_h)$ also achieves a last-iterate convergence rate of $\mathrm{NEGap}(\mu^k, \nu^k) = \widetilde{\mathcal{O}}((XA + YB) H k^{-1/8})$. Compared with Theorem 5.1, when $X$ and $A$ are sufficiently large, this rate is superior by a factor of $\widetilde{\mathcal{O}}(\sqrt{X})$ but worse by $\widetilde{\mathcal{O}}(H\sqrt{A})$. When $H$ is sufficiently large, it is superior by a factor of $\widetilde{\mathcal{O}}(X^{1/4})$*

*but worse by $\widetilde{\mathcal{O}}(A)$. However, we emphasize again that in IIEFGs, using OMD with vanilla negentropy does not permit an efficient closed-form update, even though OMD operates on the entire $\Pi_{\max}$, as opposed to the virtual transition-weighted negentropy.*

**Proof Sketch of Theorem 5.1**   We postpone the complete proof of Theorem 5.1 to Appendix C. Here we provide a proof sketch of it.

We denote by $\xi^{k,\star} = (\mu^{k,\star}, \nu^{k,\star})$ the unique NE in the regularized game $f_k$ in Eq. (4) (there is only a unique NE due to the strongly convex-strongly concave nature of $f_k$). To begin with, one can see that the NE policy profile $\xi^{k,\star}$ of $f_k$ is also an approximate NE of the original game in Eq. (3). This enables to bound $\mathrm{NEGap}(\xi^k)$ using $\mathrm{NEGap}(\xi^{k,\star})$ together with the distance between $\xi^k$ and $\xi^{k,\star}$ weighted by the virtual transitions as bellow:

$$\mathrm{NEGap}(\xi^k) \lesssim \mathrm{NEGap}(\xi^{k,\star})$$
$$+ X \left\| p^x \left( \mu^k - \mu^{k,\star} \right) \right\|_1 + Y \left\| p^y \left( \nu^k - \nu^{k,\star} \right) \right\|_1, \quad (9)$$

where $\mathrm{NEGap}(\xi^{k,\star})$ can be controlled by Lemma C.1. Due to the constructed virtual transition $p^x$ and $p^y$, the second and the third term in Eq. (9) are the $\ell_1$-norm of the difference between the probability measures over infoset-action spaces, which thus turn out to be bounded by $\mathcal{O}(\sqrt{\mathrm{KL}\left( p^x \mu^{k,\star}, p^x \mu^k \right)})$ and $\mathcal{O}(\sqrt{\mathrm{KL}\left( p^y \nu^{k,\star}, p^y \nu^k \right)})$ by Pinsker's inequality. Also, due to the constructed virtual transition-weighted negentropy $\psi$, one can deduce that $\mathrm{KL}\left( p^x \mu^{k,\star}, p^x \mu^k \right) = D_\psi(\mu^{k,\star}, \mu^k)$ (and similarly on the min-player side).

To bound the Bregman divergence $D_\psi(\mu^{k,\star}, \mu^k)$, we show that in each episode $k$, the product policy $\xi^k := (\mu^k, \nu^k)$ generated by the algorithm will approach $\xi^{k,\star}$ close enough by proving that $D_\psi(\xi^{k,\star}, \xi^k)$ is an (approximate) contraction mapping. In particular, we have

$$D_\psi\left( \xi^{k+1,\star}, \xi^{k+1} \right) \lesssim (1 - \eta_k \varepsilon_k) D_\psi\left( \xi^{k,\star}, \xi^k \right)$$
$$+ \eta_k^2 (X + Y) \tau_k + \eta_k^2 \left( X^2 A + Y^2 B \right)$$
$$+ \eta_k \rho_k + \eta_k \sigma_k + \eta_k^2 \varepsilon_k^2 H^2 (XA + YB) + \omega_k. \quad (10)$$

Please see Appendix C.2 for the detailed definitions of $\tau_k$, $\rho_k$, $\sigma_k$ and $\omega_k$.

Expanding the above recursion, we can bound $D_\psi\left( \xi^{k+1,\star}, \xi^{k+1} \right)$ as

$$D_\psi\left( \xi^{k+1,\star}, \xi^{k+1} \right) \lesssim \underbrace{\sum_{i=1}^k w_k^i \eta_i \rho_i}_{\textbf{Term 1}} + \underbrace{\sum_{i=1}^k w_k^i \eta_i \sigma_i}_{\textbf{Term 2}}$$
$$+ \underbrace{(XA + YB) \sum_{i=1}^k w_k^i (\eta_i \varepsilon_i)^2}_{\textbf{Term 3}} + \underbrace{(X + Y) \sum_{i=1}^k w_k^i \eta_i^2 \tau_i}_{\textbf{Term 4}}$$

$$+ \underbrace{\sum_{i=1}^k w_k^i \eta_i^2 \left( X^2 A + Y^2 B \right)}_{\textbf{Term 5}} + \underbrace{\sum_{i=1}^k w_k^i \omega_i}_{\textbf{Term 6}}, \quad (11)$$

where $w_k^i = \prod_{j=i+1}^k \left( 1 - \eta_j \varepsilon_j \right)$ is the contraction coefficient. Intuitively, **Term 1** and **Term 2** represent the underestimation and overestimation errors of the loss estimator, respectively. **Term 3** through **Term 5** arise from bounding the stability term of OMD. Additionally, **Term 6** is due to the variation of $f_k$ (and hence $\xi^{k,\star}$) in each episode. Then we bound each of the above terms in by Lemma C.3 - Lemma C.8 in Appendix C.2.

Finally, the proof can be concluded by substituting Eq. (11) and the upper bound of $\mathrm{NEGap}(\xi^{k,\star})$ into Eq. (9).

**Remark 5.5.** *Our analysis scheme is inspired by Cai et al. (2023) to bound the last-iterate convergence of learning matrix games with bandit feedback. However, we also remark that a straightforward application of their analysis will not address our problem of learning IIEFGs with bandit feedback, since we leverage a different regularizer and a new virtual transition $p^x$ computed by Algorithm 2. This serves as a core ingredient of the analysis when deriving the contraction in Eq. (10) and when bounding **Term 6**—one of the leading term in the final bound of the convergence rate. Besides, compared with the analysis of Cai et al. (2023), the additional **Term 5** in Eq. (11) comes from the fact that we establish a refined analysis in the case of IIEFGs to further sharpen the dependence on $X$ and $A$ (as well as $Y$ and $B$) of the final convergence rate.*

### 5.2. Lower Bound of Last-iterate Convergence

By leveraging existing regret lower bounds for learning in IIEFGs with bandit feedback (e.g., Theorem 6 of Bai et al. (2022) and Theorem 3.1 of Fiegel et al. (2023)), one can directly obtain a lower bound for learning IIEFGs that guarantees last-iterate convergence under bandit feedback. For completeness, we formalize this result as the following theorem.

**Theorem 5.6.** *For any algorithm Alg that both players adopt to generate policy profile $(\mu^k, \nu^k)$ and is uncoupled between both players, there exists an IIEFG instance such that the lower bound of the last-iterate convergence of learning this IIEFG in the bandit-feedback setting satisfies $\mathrm{NEGap}(\mu^k, \nu^k) = \Omega(\sqrt{XA + YB}k^{-1/2})$, when $k \geq \max(XA, YB)$.*

**Remark 5.7.** *Compared with the lower bound of the convergence rate above, the upper bound in Theorem 5.1 is loose by a factor of $\widetilde{\mathcal{O}}((X + Y)k^{3/8})$ (for large enough $X$, $Y$, $A$ and $B$). We believe one of the promising approaches to improve the upper bound of the convergence rate might be using the optimistic OMD/FTRL, which utilizes accelerated techniques from the optimization perspective and is*

*typically used to achieve the $\widetilde{\mathcal{O}}(1/k)$ convergence rate for learning IIEFGs with last-iterate convergence in the full-information setting. One of the main difficulties of using optimistic OMD/FTRL in conjunction with the regularization technique to achieve a faster convergence rate in the bandit feedback setting is that the loss estimator constructed in the bandit feedback setting (either unbiased or optimistically biased) would have undesirably large variance, rendering the stability of optimistic OMD/FTRL hard to be controlled even in the special case of learning matrix games. We leave the possible improvement of our convergence upper bound as our future study.*

## 6. Conclustion

In this work, we take the first step toward developing an algorithm that learns an approximate NE of IIEFGs in the bandit feedback setting with finite-time last-iterate convergence. Our algorithm operates in an entirely uncoupled manner between the two players involved, requiring no coordination, communication, or shared randomness. We prove that our algorithm achieves last-iterate convergence with a rate of $\widetilde{\mathcal{O}}(k^{-1/8})$ with high probability, and a rate of $\widetilde{\mathcal{O}}(k^{-1/6})$ in expectation. Furthermore, empirical evaluations on various IIEFG instances show the comparative advantage of our algorithm over baseline methods. A noteworthy open problem is closing the gap between the established upper and lower bounds for convergence, which remains unresolved even for the special case of learning matrix games with a last-iterate convergence guarantee in the bandit feedback setting. We leave the investigation of this for our future research endeavors.

## Impact Statement

This paper presents work whose goal is to advance the field of Machine Learning. There are many potential societal consequences of our work, none of which we feel must be specifically highlighted here.

## Acknowledgements

The corresponding author Shuai Li is partly supported by the Guangdong Provincial Key Laboratory of Mathematical Foundations for Artificial Intelligence (2023B1212010001). Baoxiang Wang is partially supported by the National Natural Science Foundation of China (72394361, 62106213) and an extended support project from the Shenzhen Science and Technology Program.

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

## A. Computation

As mentioned in Section 4.2, in practice, it suffices to consider the following relaxed optimization problem of Eq. (2), which performs the update of OMD over $\Pi_{\max}$ instead of $\Pi_{\max}^{k+1}$:

$$\mu^{k+1} = \arg\min_{\mu \in \Pi_{\max}} \eta_k \langle \mu, \widehat{\ell}^k \rangle + D_\psi(\mu, \mu^k). \tag{12}$$

Then, to prevent the loss estimates contributed by the entropy regularization from being prohibitively large, it only remains to clip the part of the entropy regularization:

$$\widehat{\ell}_h^{k+1}(x_h, a_h) = \frac{\mathbb{I}_h^{k+1}\{x_h, a_h\}}{\mu_{1:h}^{k+1}(x_h, a_h) + \gamma_k}(1 - r_h^{k+1}) + \varepsilon_{k+1} \cdot p_{1:h}^x(x_n) \log\left(\max\left\{\left[p^x \mu^{k+1}\right](x_h, a_h), \zeta^{k+1}\right\}\right), \tag{13}$$

where $\zeta^{k+1} > 0$ is the clipping threshold.

To update the policy in Eq. (12), from the proof of Proposition F.2 of Fiegel et al. (2023), one can see that the virtual transition-weighted negentropy $\psi_{p^x}(\mu) = \sum_{h,x_h,a_h} \left[p^x \mu\right](x_h, a_h) \log\left(\left[p^x \mu\right](x_h, a_h)\right)$ is equivalent to the following special dilated negentropy:

$$\psi'_k(\mu) = \sum_{h,x_h,a_h} \frac{\mu_{1:h}(x_h, a_h)}{\kappa_k(x_h)} \log\left(\frac{\mu_{1:h}(x_h, a_h)}{\mu_{1:h}(x_h)}\right) + \sum_{h,x_h,a_h} \left[p^x \mu\right](x_h, a_h) \log\left(p_{1:h}^x(x_h)\right). \tag{14}$$

where

$$\kappa_k(x_h) = \eta_k / ((H - h + 1) p_{1:h}^x(x_h)). \tag{15}$$

Since the second term in Eq. (14) is a linear term in $\mu$, the virtual transition-weighted negentropy $\psi$ and this special dilated negentropy $\psi'$ will induce the same Bregman divergence. Hence, $\mu^{k+1}$ computed in Eq. (12) is the same as it computed by the following:

$$\mu^{k+1} = \arg\min_{\mu \in \Pi_{\max}} \eta_k \langle \mu, \widehat{\ell}^k \rangle + D_{\psi'_k}(\mu, \mu^k). \tag{16}$$

We conclude the discussion here by demonstrating Algorithm 3, which provides a closed-form update of Eq. (16). The correctness of Algorithm 3 is guaranteed in Proposition A.1.

---

**Algorithm 3** Closed-form Solution of Eq. (16)

---

1: **Input:** Tree-like structure of $\mathcal{X} \times \mathcal{A}$, fixed learning rates $\eta$, virtual transition $p^\star$ and clipped loss estimator $\widehat{\ell}^k$.
2: **Initialization:** For all $x_H$ in $\mathcal{X}_H$, initialize $Z^k(x_{H+1}) = 1$. Set adaptive learning rates $\kappa_k$ according to Eq. (15).
3: **for** $h = H$ to $1$ **do**
4:     **for** $x_h$ in $\mathcal{X}_h$ **do**
5:         Compute $J_h^k(x_h, a_h) = -\kappa_k(x_h)\widehat{\ell}_h^k(x_h, a_h) + \sum_{x_{h+1} \in C(x_h, a_h)} \frac{\kappa_k(x_h)}{\kappa_k(x_{h+1})} \log Z_{h+1}^k(x_{h+1})$.
6:         Compute $Z_h^k(x_h) = \sum_{a_h \in \mathcal{A}} \mu_h^\star(a_h|x_h) \exp\left(J_h^k(x_h, a_h)\right)$.
7:         **for** $a_h$ in $\mathcal{A}$ **do**
8:             Compute $\mu_h^{k+1}(a_h|x_h) = \mu_h^\star(a_h|x_h) \exp\left(J_h^k(x_h, a_h) - \log Z_h^k(x_h)\right)$.
9:         **end for**
10:     **end for**
11: **end for**

---

**Proposition A.1.** *The solution to the update in Eq. (16) satisfies*

$$\mu_h^{k+1}(a_h|x_h) = \mu_h^k(a_h|x_h) \exp\left\{-\kappa_k(x_h)\widehat{\ell}_h^k(x_h, a_h) + \sum_{x_{h+1} \in C(x_h, a_h)} \frac{\kappa_k(x_h)}{\kappa_k(x_{h+1})} \log Z_{h+1}^k(x_{h+1}) - \log Z_h^k(x_h)\right\},$$

*where*

$$Z_h^k(x_h) = \sum_{a_h \in \mathcal{A}} \mu_h^k(a_h|x_h) \exp\left\{ -\kappa_k(x_h)\hat{\ell}_h^k(x_h, a_h) + \sum_{x_{h+1} \in C(x_h, a_h)} \frac{\kappa_k(x_h)}{\kappa_k(x_{h+1})} \log Z_{h+1}^k(x_{h+1}) \right\}, \qquad (17)$$

*and for notational convenience, we define that* $\forall (x_H, a_H) \in \mathcal{X}_H \times \mathcal{A}$*, it has a unique descendant* $x_{H+1}$ *such that* $Z_{H+1}^k(x_{H+1}) = 1$.

*Proof.* First note that

$$\left\langle \mu, \widehat{\ell}^k \right\rangle + D_{\psi'}(\mu, \mu^k) = \sum_{h=1}^{H} \sum_{(x_h, a_h) \in \mathcal{X}_h \times \mathcal{A}} \mu_{1:h}(x_h, a_h) \left[ \widehat{\ell}_h^k(x_h, a_h) + \frac{1}{\kappa_k(x_h)} \log \frac{\mu_h(a_h|x_h)}{\mu_h^k(a_h|x_h)} \right]$$

$$= \sum_{h=1}^{H} \sum_{x_h \in \mathcal{X}_h} \mu_{1:h-1}(x_h) \left[ \left\langle \mu_h(\cdot|x_h), \widehat{\ell}_h^k(x_h, \cdot) \right\rangle + \frac{\text{KL}\left(\mu_h(\cdot|x_h), \mu_h^k(\cdot|x_h)\right)}{\kappa_k(x_h)} \right]. \qquad (18)$$

We now prove the proposition via backward induction over $h = H, \ldots, 1$.

When $h = H$, for any $x_H \in \mathcal{X}_H$, Eq. (18) shows that

$$\mu_H^{k+1}(a_H|x_H) = \mu_H^k(a_H|x_H) \exp\left\{ -\kappa_k(x_h)\widehat{\ell}_H^k(x_H, a_H) - \log Z_H^k(x_H) \right\},$$

where $Z_H^k(x_H) = \sum_{a_H \in \mathcal{A}} \mu_H^k(a_H|x_H) \exp\{ -\kappa_k(x_h)\widehat{\ell}_H^k(x_H, a_H) \}$ is a normalization factor.

Fix some $h \in [H]$. Now suppose the induction hypothesis holds from step $h+1$ to $H$ and consider the $h$-th step. Using the induction hypothesis, one can see that Eq. (18) can be rewritten as

$$\sum_{h'=1}^{H} \sum_{(x_{h'}, a_{h'}) \in \mathcal{X}_{h'} \times \mathcal{A}} \mu_{1:h'}(x_{h'}, a_{h'}) \left[ \widehat{\ell}_{h'}^k(x_{h'}, a_{h'}) + \frac{1}{\kappa_k(x_{h'})} \log \frac{\mu_{h'}(a_{h'}|x_{h'})}{\mu_{h'}^k(a_{h'}|x_{h'})} \right]$$

$$= \sum_{h'=1}^{H} \sum_{x_{h'} \in \mathcal{X}_{h'}} \mu_{1:h'-1}(x_{h'}) \left[ \left\langle \mu_{h'}(\cdot|x_{h'}), \widehat{\ell}_{h'}^k(x_{h'}, \cdot) \right\rangle + \frac{\text{KL}\left(\mu_{h'}(\cdot|x_{h'}), \mu_{h'}^k(\cdot|x_{h'})\right)}{\kappa_k(x_{h'})} \right]$$

$$= \sum_{h'=1}^{h} \sum_{x_{h'} \in \mathcal{X}_{h'}} \mu_{1:h'-1}(x_{h'}) \left[ \left\langle \mu_{h'}(\cdot|x_{h'}), \widehat{\ell}_{h'}^k(x_{h'}, \cdot) \right\rangle + \frac{\text{KL}\left(\mu_{h'}(\cdot|x_{h'}), \mu_{h'}^k(\cdot|x_{h'})\right)}{\kappa_k(x_{h'})} \right]$$

$$+ \sum_{h'=h+1}^{H} \left[ \sum_{x_{h'+1} \in \mathcal{X}_{h'+1}} \frac{\mu_{1:h'}(x_{h'+1})}{\kappa_k(x_{h'+1})} \log Z_{h'+1}^k(x_{h'+1}) - \sum_{x_{h'} \in \mathcal{X}_{h'}} \frac{\mu_{1:h'-1}(x_{h'})}{\kappa_k(x_{h'})} \log Z_{h'}^k(x_{h'}) \right]$$

$$= \sum_{h'=1}^{h} \sum_{x_{h'} \in \mathcal{X}_{h'}} \mu_{1:h'-1}(x_{h'}) \left[ \left\langle \mu_{h'}(\cdot|x_{h'}), \widehat{\ell}_{h'}^k(x_{h'}, \cdot) \right\rangle + \frac{\text{KL}\left(\mu_{h'}(\cdot|x_{h'}), \mu_{h'}^k(\cdot|x_{h'})\right)}{\kappa_k(x_{h'})} \right]$$

$$- \sum_{x_{h+1} \in \mathcal{X}_{h+1}} \frac{\mu_{1:h}(x_{h+1})}{\kappa_k(x_{h+1})} \log Z_{h+1}^k(x_{h+1})$$

$$= \sum_{h'=1}^{h-1} \sum_{x_{h'} \in \mathcal{X}_{h'}} \mu_{1:h'-1}(x_{h'}) \left[ \left\langle \mu_{h'}(\cdot|x_{h'}), \widehat{\ell}_{h'}^k(x_{h'}, \cdot) \right\rangle + \frac{\text{KL}\left(\mu_{h'}(\cdot|x_{h'}), \mu_{h'}^k(\cdot|x_{h'})\right)}{\kappa_k(x_{h'})} \right]$$

$$+ \sum_{x_h \in \mathcal{X}_h} \mu_{1:h-1}(x_h) \left[ \underbrace{\left\langle \mu_h(\cdot|x_h), \widehat{\ell}_h^k(x_h, \cdot) - \sum_{x_{h+1} \in C(x_h, \cdot)} \frac{\log Z_{h+1}^k(x_{h+1})}{\kappa_k(x_{h+1})} \right\rangle}_{\heartsuit} + \frac{\text{KL}\left(\mu_h(\cdot|x_h), \mu_h^k(\cdot|x_h)\right)}{\kappa_k(x_h)} \right].$$

By minimizing ($\heartsuit$), one can derive that

$$\mu_h^{k+1}(a_h|x_h) = \mu_h^k(a_h|x_h) \exp\left\{ -\kappa_k(x_h)\hat{\ell}_h^k(x_h,a_h) + \sum_{x_{h+1}\in C(x_h,a_h)} \frac{\kappa_k(x_h)}{\kappa_k(x_{h+1})} \log Z_{h+1}^k(x_{h+1}) - \log Z_h^k(x_h) \right\},$$

where

$$Z_h^k(x_h) = \sum_{a_h\in\mathcal{A}} \mu_h^k(a_h|x_h) \exp\left\{ -\kappa_k(x_h)\hat{\ell}_h^k(x_h,a_h) + \sum_{x_{h+1}\in C(x_h,a_h)} \frac{\kappa_k(x_h)}{\kappa_k(x_{h+1})} \log Z_{h+1}^k(x_{h+1}) \right\}.$$

The proof is thus concluded. $\qquad\square$

## B. More Discussions on Virtual Transition Probabilities

### B.1. Illustration on the Failure of Using Uniform Virtual Transition

In this section, we demonstrate why a uniform virtual transition is insufficient for achieving meaningful last-iterate convergence rates.

Consider an IIEFG instance where there is only one action $a$ and $H = 4$. Each infoset $x$ in the game tree of this instance satisfies $|C(x,a)| = 2$ except for infoset $x_{2,1}$, which is such that $|C(x_{2,1},a)| = n$ with some $n \geq 2$. Now, suppose the uniform distribution $p$ is used as a virtual transition over infoset-action spaces. Then for all the descendants $\{x_{4,i}\}_{i=1}^{2n}$ on step $h = 4$ of infoset $x_{2,1}$, one can see that $p_{1:H}(x_{H,i}) = \frac{1}{2} \cdot \frac{1}{n} \cdot \frac{1}{2} = \frac{1}{4n}$, while there are only $X = 9 + 3n$ infosets in total. Thus, it will happen that $p_{1:H}(x_{H,i}) < \frac{1}{X}$ when $n > 9$.

Actually, one can easily construct an IIEFG instance such that $\min_{x_H\in\mathcal{X}_H} p_{1:H}(x_H) \leq \mathcal{O}(\frac{1}{n^m})$ and $X = \mathcal{O}(mn + c)$ with $c$ as a parameter that depends on $m$ but not $n$ for uniform virtual transition $p$. Therefore, when using uniform distribution $p$ as a virtual transition, $\max_{x_H\in\mathcal{X}_H} 1/p_{1:H}(x_H)$ might be prohibitively large and lead to a convergence rate with much worse dependence on $X$ than the virtual transition constructed in our Algorithm 2.

### B.2. Balanced Effects of the Proposed Virtual Transition Probability

**Lemma B.1.** *For any $h \in [H]$ and $x_h \in \mathcal{X}_h$, the constructed virtual transition $p^x$ guarantees that $1/p_{1:h}^x(x_h) \leq X$.*

*Proof.* Clearly, $p_{1:h}^x(\cdot)$ is minimzed at $h = H$ for some $x_H \in \mathcal{X}_H$ by the definition of virtual transition. By the construction of $p_{1:h}^x(\cdot)$ in Algorithm 2, one can deduce that $\forall x_H \in \mathcal{X}_H$, it holds that (understanding $\{(x_h,a_h)\}_{h\in[H-1]}$ as the unique trajectory leading to $x_H$ below)

$$\begin{aligned}
p_{1:H}^x(x_H) &= q[x_H] \\
&= q[x_{H-1}] \cdot \frac{c[x_H]}{\sum_{x_H'\in C(x_{H-1},a_{H-1})} c[x_H']} \\
&= q[x_{H-2}] \cdot \frac{c[x_{H-1}]}{\sum_{x_{H-1}'\in C(x_{H-2},a_{H-2})} c[x_{H-1}']} \cdot \frac{c[x_H]}{\sum_{x_H'\in C(x_{H-1},a_{H-1})} c[x_H']} \\
&= q[x_{H-2}] \cdot \frac{c[x_{H-1}]}{\sum_{x_{H-1}'\in C(x_{H-2},a_{H-2})} c[x_{H-1}']} \cdot \frac{c[x_H]}{d[x_{H-1},a_{H-1}]} \\
&\overset{(i)}{\geq} q[x_{H-2}] \cdot \frac{c[x_{H-1}]}{\sum_{x_{H-1}'\in C(x_{H-2},a_{H-2})} c[x_{H-1}']} \cdot \frac{c[x_H]}{c[x_{H-1}]} \\
&= q[x_{H-2}] \cdot \frac{c[x_H]}{\sum_{x_{H-1}'\in C(x_{H-2},a_{H-2})} c[x_{H-1}']} \\
&\geq \dots \\
&\geq \frac{c[x_H]}{\sum_{x_1\in\mathcal{X}_1} c[x_1]}
\end{aligned}$$

$$\geq \frac{c\,[x_H]}{X_H}$$
$$\geq \frac{c\,[x_H]}{X}$$
$$= \frac{1}{X},$$

where $c[\cdot]$, $q[\cdot]$, and $d[\cdot,\cdot]$ are defined in our Algorithm 2; and $(i)$ is due to $c[x_{H-1}] = \max_{a \in \mathcal{A}} d[x_{H-1}, a] \geq d[x_{H-1}, a_{H-1}]$.
□

The property shown in this lemma of our constructed virtual transition $p^x$ serves as a key ingredient in the analysis (say, when bounding our **Term 4** as well as **Term 6** and when establishing the final convergence upper bound of the NE gap in the proof of Theorem 5.1) as we shall see.

## C. Proof of Theorem 5.1

*Proof of Theorem 5.1.* Throughout the proof, since Theorem 5.1 holds for all $k$ as fixed constants, we assume $k \geq k_0 = \left( \frac{24}{1-\alpha_\eta-\alpha_\epsilon} \ln \left( \frac{12}{1-\alpha_\eta-\alpha_\epsilon} \right) \right)^{\frac{1}{1-\alpha_\eta-\alpha_\epsilon}} = (96\ln(48))^4$. In what follows, we denote by $m_k^x = \min_{h,x_h,a_h} \left[ p^x \mu^k \right](x_h, a_h)$ and $m_k^y = \min_{h,y_h,b_h} \left[ p^y \nu^k \right](y_h, b_h)$.

First, notice that $\text{NEGap}(\xi_k)$ can be translated into the summation of $\text{NEGap}(\xi^{k,\star})$ and the distance of $\xi^k$ to $\xi^{k,\star}$ as follows:

$$
\begin{aligned}
&\text{NEGap}\left(\xi^k\right)\\
&= \sup_{\mu \in \Pi_{\max}, \nu \in \Pi_{\min}} f\left(\mu^k, \nu\right) - f\left(\mu, \nu^k\right)\\
&= \sup_{\mu \in \Pi_{\max}, \nu \in \Pi_{\min}} f\left(\mu^{k,\star}, \nu\right) - f\left(\mu^{k,\star}, \nu\right) + f\left(\mu^k, \nu\right) - f\left(\mu, \nu^k\right) + f\left(\mu, \nu^{k,\star}\right) - f\left(\mu, \nu^{k,\star}\right)\\
&\leq \sup_{\mu \in \Pi_{\max}, \nu \in \Pi_{\min}} \text{NEGap}\left(\xi^{k,\star}\right) + \left(\mu^k - \mu^{k,\star}\right)^\top \boldsymbol{G}\nu + \mu^\top \boldsymbol{G}\left(\nu^{k,\star} - \nu^k\right)\\
&\overset{(i)}{\leq} \sup_{\mu \in \Pi_{\max}, \nu \in \Pi_{\min}} \text{NEGap}\left(\xi^{k,\star}\right) + \left\langle p^x\left(\mu^k - \mu^{k,\star}\right), \boldsymbol{G}\nu/p^x \right\rangle + \left\langle p^y\left(\nu^k - \nu^{k,\star}\right), \boldsymbol{G}^\top\mu/p^y \right\rangle\\
&\leq \sup_{\mu \in \Pi_{\max}, \nu \in \Pi_{\min}} \text{NEGap}\left(\xi^{k,\star}\right) + \left\| p^x\left(\mu^k - \mu^{k,\star}\right)\right\|_1 \left\|\boldsymbol{G}\nu/p^x\right\|_\infty + \left\| p^y\left(\nu^k - \nu^{k,\star}\right)\right\|_1 \left\|\boldsymbol{G}^\top\mu/p^y\right\|_\infty\\
&\overset{(ii)}{\leq} \text{NEGap}\left(\xi^{k,\star}\right) + X\left\| p^x\left(\mu^k - \mu^{k,\star}\right)\right\|_1 + Y\left\| p^y\left(\nu^k - \nu^{k,\star}\right)\right\|_1\\
&\overset{(iii)}{\leq} \varepsilon_k H\left(\ln(XA) + \ln(YB)\right) + \mathcal{O}\left(\frac{XAH}{k} + \frac{YBH}{k} + X\sqrt{\text{KL}\left(p^x\mu^{k,\star}, p^x\mu^k\right)} + Y\sqrt{\text{KL}\left(p^y\nu^{k,\star}, p^y\nu^k\right)}\right)\\
&\leq \varepsilon_k H\left(\ln(XA) + \ln(YB)\right) + \mathcal{O}\left(\frac{XAH}{k} + \frac{YBH}{k} + (X+Y)\sqrt{\text{KL}\left(p^x\mu^{k,\star}, p^x\mu^k\right) + \text{KL}\left(p^y\nu^{k,\star}, p^y\nu^k\right)}\right)\\
&\overset{(iv)}{\leq} \varepsilon_k H\left(\ln(XA) + \ln(YB)\right) + \mathcal{O}\left(\frac{XAH}{k} + \frac{YBH}{k} + (X+Y)\sqrt{\text{KL}\left(p^z\xi^{k,\star}, p^z\xi^k\right)}\right)\\
&\overset{(v)}{\leq} \varepsilon_k H\left(\ln(XA) + \ln(YB)\right) + \mathcal{O}\left(\frac{XAH}{k} + \frac{YBH}{k} + (X+Y)\sqrt{D_\psi\left(\xi^{k,\star}, \xi^k\right)}\right),
\end{aligned}
\tag{19}
$$

where in $(i)$ $\boldsymbol{G}\nu/p^x \in \mathbb{R}^{XA}$ is defined such that $(\boldsymbol{G}\nu/p^x)[(x_h, a_h)] = (\boldsymbol{G}\nu)[(x_h, a_h)]/p_{1:h}^x(x_h)$ and similarly for $\boldsymbol{G}^\top\mu/p^y$; $(ii)$ is by Lemma B.1; $(iii)$ is by Lemma C.1 and Pinsker's inequality; in $(iv)$ we denote by $p^z = (p^x, p^y)$; and $(v)$ follows from the definition of the virtual transition-weighted negentropy regularizer in Eq. (6).

We proceed to bound $D_\psi\left(\xi^{k,\star}, \xi^k\right)$ in the above display. To this end, putting Lemma C.2, C.3, C.4, C.5, C.6, C.7, and C.8 together leads to

$$D_\psi\left(\xi^{k+1,\star}, \xi^{k+1}\right)$$

$$\lesssim (XA + YB) \ln(k) k^{-\alpha_\gamma + \alpha_\varepsilon} + k^{-\frac{\alpha_k}{2} + \frac{\alpha_\varepsilon}{2}} \log\left(k^2/\delta\right) + k^{-\alpha_\eta + \alpha_\gamma} \log(k^2/\delta)$$
$$+ \left(XA(\log^2 m_k^x) + YB(\log^2 m_k^y)\right) k^{-\alpha_\eta - \alpha_\varepsilon} + k^{\alpha_\gamma - 2\alpha_\eta}(X + Y) \log\left(1/\delta\right) + \left(X^2 A + Y^2 B\right) k^{-\alpha_\eta + \alpha_\varepsilon}$$
$$+ (X + Y)^{\frac{1}{2}} \log^2\left(1/(m_k^x m_k^y)\right) \log(k) k^{-\min\{1, (3 - \alpha_\varepsilon)/2\} + \alpha_\eta + \alpha_\epsilon}$$

$$\leq \mathcal{O}\left((XA + YB) \ln(k) k^{-\alpha_\gamma + \alpha_\varepsilon} + k^{-\frac{\alpha_\eta}{2} + \frac{\alpha_\varepsilon}{2}} \log\left(\frac{k^2}{\delta}\right) + k^{-\alpha_\eta + \alpha_\gamma} \log\left(\frac{k^2}{\delta}\right)\right.$$
$$+ \left(XA\left(\log X + H \log\left(Ak\right)^2 + YB\left(\log Y + H \log\left(Bk\right)\right)^2\right)\right) k^{-\alpha_\eta - \alpha_\varepsilon}$$
$$+ k^{\alpha_\gamma - 2\alpha_\eta}(X + Y) \log\left(\frac{1}{\delta}\right) + (X^2 A + Y^2 B) k^{-\alpha_\eta + \alpha_\epsilon}$$
$$+ (X + Y)^{\frac{1}{2}} \left(H \log\left(Ak\right) + H \log\left(Bk\right)\right) \left(\log X + H \log\left(k\right) + \log Y + H \log\left(Bk\right)\right)$$
$$\left.\cdot \log(k) k^{-\min\{1, (3 - \alpha_\varepsilon)/2\} + \alpha_\eta + \alpha_\epsilon}\right)$$

$$\leq \mathcal{O}\left(\left[k^{-\frac{1}{4}}(XA + YB) + k^{-\frac{1}{4}} + k^{-\frac{1}{4}} + (XA + YB)H^2 k^{-\frac{3}{4}} + (X + Y)k^{-\frac{7}{8}} + \left(X^2 A + Y^2 B\right) k^{-\frac{1}{2}}\right.\right.$$
$$\left.\left.+ (X + Y)^{\frac{1}{2}} H^2 K^{-\frac{1}{4}}\right] \left(\log^2\left(XAk/\delta\right) + \log^2\left(YBk/\delta\right)\right) \log(K)\right), \tag{20}$$

where the second inequality is by the fact that $\log(1/m_k^x) \leq (\log X + H \log\left(Ak\right))$ and $\log(1/m_k^y) \leq (\log Y + H \log\left(Bk\right))$.

Substituting Eq. (20) into Eq. (19) shows that

$$\text{NEGap}(\mu^k, \nu^k)$$
$$= \mathcal{O}\left((X + Y)\left[k^{-\frac{1}{8}}(XA + YB)^{\frac{1}{2}} + (XA + YB)^{\frac{1}{2}} H k^{-\frac{3}{8}} + \left(X^2 A + Y^2 B\right)^{\frac{1}{2}} k^{-\frac{1}{4}} + (X + Y)^{\frac{1}{4}} H K^{-\frac{1}{8}}\right]\right.$$
$$\left.\cdot \left(\log\left(XAk/\delta\right) + \log\left(YBk/\delta\right)\right) \log^{\frac{1}{2}}(k) + k^{-\frac{1}{8}} H(\ln(XA) + \ln(YB)) + \frac{XAB}{k} + \frac{YBH}{k}\right)$$
$$= \widetilde{\mathcal{O}}\left((X + Y)\left[k^{-\frac{1}{8}}(XA + YB)^{\frac{1}{2}} + (XA + YB)^{\frac{1}{2}} H K^{-\frac{3}{8}} + \left(X^2 A + Y^2 B\right)^{\frac{1}{2}} k^{-\frac{1}{4}} + (X + Y)^{\frac{1}{4}} H k^{-\frac{1}{8}}\right]\right.$$
$$\left.+ \frac{(XAH + YBH)}{k}\right)$$
$$= \widetilde{\mathcal{O}}\left((X + Y)k^{-\frac{1}{8}}\left[(XA + YB)^{\frac{1}{2}} + (X + Y)^{\frac{1}{4}} H\right]\right),$$

where the last equality holds when $k \geqslant \max\{H^4, \left(X^2 A + Y^2 B\right)^4/(XA + YB)^4, (XA + YB)^{8/7}/(X + Y)^{10/7}\}$.

The proof is thus concluded. $\qquad\square$

## C.1. Bounding $\text{NEGap}(\xi^{k,\star})$

**Lemma C.1.** $\forall k \geqslant 1$, it holds that

$$\text{NEGap}(\xi^{k,\star}) = \mathcal{O}\left(\varepsilon_k H(\ln(XA) + \ln(YB)) + \frac{XAH}{k} + \frac{YBH}{k}\right). \tag{21}$$

*Proof.* Fix arbitrary $(\mu', \nu') \in \Pi_{\max} \times \Pi_{\min}$. Note that

$$f\left(\mu^{k,\star}, \nu'\right) - f\left(\mu', \nu^{k,\star}\right)$$
$$= f\left(\mu^{k,\star}, \nu'\right) - f\left(\mu^{k,\star}, \nu\right) + f\left(\mu^{k,\star}, \nu\right) - f\left(\mu, \nu^{k,\star}\right) + f\left(\mu, \nu^{k,\star}\right) - f\left(\mu', \nu^{k,\star}\right), \tag{22}$$

where $(\mu, \nu) \in \Pi_{\max}^k \times \Pi_{\min}^k$ is such that $\mu = \arg\min_{\widehat{\mu} \in \Pi_{\max}^k} \|\widehat{\mu} - \mu'\|_\infty$ and $\nu = \arg\min_{\widehat{\nu} \in \Pi_{\min}^k} \|\widehat{\nu} - \nu'\|_\infty$. It is clear that $f\left(\mu^{k,\star}, \nu\right) - f\left(\mu, \nu^{k,\star}\right)$ in the above display can be bounded as

$$f\left(\mu^{k,\star}, \nu\right) - f\left(\mu, \nu^{k,\star}\right)$$
$$= f\left(\mu^{k,\star}, \nu\right) - f_k\left(\mu^{k,\star}, \nu\right) + f_k\left(\mu^{k,\star}, \nu\right) - f_k\left(\mu, \nu^{k,\star}\right) + f_k\left(\mu, \nu^{k,\star}\right) - f\left(\mu, \nu^{k,\star}\right)$$

$$
\begin{aligned}
&\overset{(i)}{\leq} - \left( \varepsilon_k \psi \left( \mu^{k,\star} \right) - \varepsilon_k \psi(\nu) \right) + \left( \varepsilon_k \psi(\mu) - \varepsilon_k \psi \left( \nu^{k,\star} \right) \right) \\
&\overset{(ii)}{\leq} - \varepsilon_k \psi \left( \mu^{k,\star} \right) - \varepsilon_k \psi \left( \nu^{k,\star} \right) \\
&\leq \varepsilon_k H (\ln(XA) + \ln(YB)) ,
\end{aligned}
\tag{23}
$$

where $(i)$ is due to that $(\mu^{k,\star}, \nu^{k,\star})$ is the NE policy profile of $f_k$; and $(ii)$ follows from $\psi(\mu) \leq 0$ for all $\mu \in \Pi_{\max}$ (similarly for all $\nu \in \Pi_{\min}$).

Moreover, $f \left( \mu^{k,\star}, \nu' \right) - f \left( \mu^{k,\star}, \nu \right)$ in Eq. (22) can be bounded as

$$
\begin{aligned}
f \left( \mu^{k,\star}, \nu' \right) - f \left( \mu^{k,\star}, \nu \right) &= \left\langle \nabla_\nu f \left( \mu^{k,\star}, \nu \right), \nu' - \nu \right\rangle \\
&\leq \| \nabla_\nu f \left( \mu^{k,\star}, \nu \right) \|_1 \| \nu' - \nu \|_\infty \\
&\overset{(i)}{\leq} YB \left( 1 - \left( 1 - \frac{B-1}{Bk} \right)^H \right) \\
&\leq YB \left( 1 - \left( 1 - \frac{1}{k} \right)^H \right) \\
&\overset{(ii)}{=} \mathcal{O} \left( \frac{YBH}{k} \right) ,
\end{aligned}
\tag{24}
$$

where $(i)$ is due to Lemma 2 of Kozuno et al. (2021) and the definition of $\Pi_{\max}^k$ in Algorithm 1; and $(ii)$ follows from $(1 - 1/k)^H = \mathcal{O} \left( 1 - \binom{H}{1}/k + \binom{H}{2}/k^2 - \dots \right) = \mathcal{O}(1 - H/k)$. Similarly, we have

$$
f \left( \mu, \nu^{k,\star} \right) - f \left( \mu', \nu^{k,\star} \right) \leq \mathcal{O} \left( \frac{XAH}{k} \right) .
\tag{25}
$$

The proof is concluded by substituting Eq. (23), Eq. (24) and Eq. (25) into Eq. (22) and noticing that the above holds for all $(\mu', \nu') \in \Pi_{\max} \times \Pi_{\min}$. $\qquad\square$

## C.2. Bounding Contraction of Bregman Divergences

**Lemma C.2** (Contraction of Bregman divergences). *The Bregman divergences of the virtual transition-weighted negentropy regularizer satisfies*

$$
\begin{aligned}
&D_\psi \left( \xi^{k+1,\star}, \xi^{k+1} \right) \\
&\leq \underbrace{\sum_{i=1}^k w_k^i \eta_i \rho_i}_{\textit{Term 1}} + \underbrace{\sum_{i=1}^k w_k^i \eta_i \sigma_i}_{\textit{Term 2}} + \underbrace{XA(\log^2 m_k^x) \sum_{i=1}^k w_k^i \left( \eta_i \varepsilon_i \right)^2 + YB(\log^2 m_k^y) \sum_{i=1}^k w_k^i \left( \eta_i \varepsilon_i \right)^2}_{\textit{Term 3}} \\
&\quad + \underbrace{\sum_{i=1}^k w_k^i \eta_i^2 \left( X \underline{\tau}_i + Y \bar{\tau}_i \right)}_{\textit{Term 4}} + \underbrace{\sum_{i=1}^k w_k^i \eta_i^2 \left( X^2 A + Y^2 B \right)}_{\textit{Term 5}} + \underbrace{\sum_{i=1}^k w_k^i \omega_i}_{\textit{Term 6}} ,
\end{aligned}
\tag{26}
$$

*where $w_k^i = \prod_{j=i+1}^k (1 - \eta_j \varepsilon_j)$.*

*Proof.* We start by deriving a descent inequality for $D_\psi \left( \mu^{k+1,\star}, \mu^{k+1} \right)$ using the difference between $f_k(\mu^k, \nu^k) - f_k(\mu^{k,\star}, \nu^k)$:

$$
f_k(\mu^k, \nu^k) - f_k(\mu^{k,\star}, \nu^k) = \left( \mu^k - \mu^{k,\star} \right)^\top \boldsymbol{G} \nu^k + \varepsilon_k \left( \psi(\mu^k) - \psi(\mu^{k,\star}) \right) .
\tag{27}
$$

For the first term on the RHS of the above display, we have

$$
\left( \mu^k - \mu^{k,\star} \right)^\top \boldsymbol{G} \nu^k
$$

$$= \left(\mu^k - \mu^{k,\star}\right)^\top \left(\boldsymbol{G}\nu^k + \widehat{\ell}^k - \widehat{\ell}^k\right)$$

$$= \left(\mu^k - \mu^{k,\star}\right)^\top \widehat{\ell}^k + \left(\mu^k\right)^\top \left(\boldsymbol{G}\nu^k - \widehat{\ell}^k\right) - \left(\mu^{k,\star}\right)^\top \left(\boldsymbol{G}\nu^k - \widehat{\ell}^k\right)$$

$$= \left(\mu^k - \mu^{k,\star}\right)^\top \widehat{\ell}^k + \sum_{h,x_h,a_h} \mu_{1:h}^k (x_h, a_h) \left(\left[\boldsymbol{G}\nu^k\right](x_h, a_h) - \widehat{\ell}^k(x_h, a_h)\right)$$

$$- \sum_{h,x_h,a_h} \mu_{1:h}^{k,\star}(x_n, a_n) \left(\left[\boldsymbol{G}\nu^k\right](x_h, a_h) - \widehat{\ell}^k(x_h, a_h)\right)$$

$$= \left(\mu^k - \mu^{k,\star}\right)^\top \widehat{\ell}^k$$

$$+ \sum_{h,x_h,a_h} \mu_{1:h}^k (x_h, a_h) \left[\left[\boldsymbol{G}\nu^k\right](x_h, a_h) - \left(\frac{\mathbb{I}_h^k \{x_h, a_h\}}{\mu_{1:h}^k (x_h, a_h) + \gamma_k} \left(1 - r_h^k\right) + \varepsilon_k p_{1:h}^x (x_h) \log\left[p^x \mu^k\right](x_h, a_h)\right)\right]$$

$$- \sum_{h,x_h,a_h} \mu_{1:h}^{k,\star}(x_h, a_h) \left[\left[\boldsymbol{G}\nu^k\right](x_h, a_h) - \left(\frac{\mathbb{I}_h^k \{x_h, a_h\}}{\mu_{1:h}^k (x_h, a_h) + \gamma_k} \left(1 - r_h^k\right) + \varepsilon_k p_{1:h}^x (x_h) \log\left[p^x \mu^k\right](x_h, a_h)\right)\right].$$

$$(28)$$

For the second term of the RHS of Eq. (27), we have

$$\psi(\mu^k) - \psi(\mu^{k,\star})$$

$$= \sum_{h,x_h,a_h} \left[p^x \mu^k\right](x_h, a_h) \log\left[p^x \mu^k\right](x_h, a_h) - \sum_{h,x_h,a_h} \left[p^x \mu^{k,\star}\right](x_h, a_h) \log\left[p^x \mu^{k,\star}\right](x_h, a_h)$$

$$= \sum_{h,x_h,a_h} \left(\left[p^x \mu^k\right](x_h, a_h) - \left[p^x \mu^{k,\star}\right](x_h, a_h)\right) \log\left[p^x \mu^k\right](x_h, a_h)$$

$$- \sum_{h,x_h,a_h} \left[p^x \mu^{k,\star}\right](x_h, a_h) \left(\log\left[p^x \mu^{k,\star}\right](x_h, a_h) - \log\left[p^x \mu^k\right](x_h, a_h)\right)$$

$$= \sum_{h,x_h,a_h} \left(\left[p^x \mu^k\right](x_h, a_h) - \left[p^x \mu^{k,\star}\right](x_h, a_h)\right) \log\left[p^x \mu^k\right](x_h, a_h) - D_\psi(\mu^{k,\star}, \mu^k),$$

$$(29)$$

where the last equality again follows from the definition of the virtual transition-weighted negentropy regularizer in Eq. (6).

Substituting Eq. (28) and Eq. (29) into Eq. (27) leads to

$$f_k(\mu^k, \nu^k) - f_k(\mu^{k,\star}, \nu^k)$$

$$= \left(\mu^k - \mu^{k,\star}\right)^\top \widehat{\ell}^k + \underbrace{\sum_{h,x_h,a_h} \mu_{1:h}^k (x_h, a_h) \left[\left[\boldsymbol{G}\nu^k\right](x_h, a_h) - \frac{\mathbb{I}_h^k \{x_h, a_h\}}{\mu_{1:h}^k (x_h, a_h) + \gamma_k} \left(1 - r_h^k\right)\right]}_{=: \underline{\rho}_k}$$

$$\underbrace{- \sum_{h,x_h,a_h} \mu_{1:h}^{k,\star}(x_h, a_h) \left[\left[\boldsymbol{G}\nu^k\right](x_h, a_h) - \frac{\mathbb{I}_h^k \{x_h, a_h\}}{\mu_{1:h}^k (x_h, a_h) + \gamma_k} \left(1 - r_h^k\right)\right]}_{=: \underline{\sigma}_k} - \varepsilon_k D_\psi(\mu^{k,\star}, \mu^k)$$

$$\overset{(i)}{\leq} \frac{1}{\eta_k} \left(D_\psi(\mu^{k,\star}, \mu^k) - D_\psi(\mu^{k,\star}, \mu^{k+1})\right) - \varepsilon_k D_\psi(\mu^{k,\star}, \mu^k) + \underline{\rho}_k + \underline{\sigma}_k$$

$$+ \sum_{h,x_h,a_h} \eta_k \left(\frac{1}{p_{1:h}^x (x_h)} \mu_{1:h}^k (x_h, a_h) \widehat{\ell}_h^k (x_h, a_h)^2 + \varepsilon_k^2 \log^2 \left[p^x \mu^k\right](x_h, a_h)\right)$$

$$\overset{(ii)}{\leq} \frac{(1 - \eta_k \varepsilon_k) D_\psi\left(\mu^{k,\star}, \mu^k\right) - D_\psi\left(\mu^{k,\star}, \mu^{k+1}\right)}{\eta_k} + \underline{\rho}_k + \underline{\sigma}_k$$

$$+ \sum_{h,x_h,a_h} \eta_k \left(\frac{1}{p_{1:h}^x (x_h)} \frac{\mathbb{I}_h^k \{x_h, a_h\}}{\mu_{1:h}^k (x_h, a_h) + \gamma_k} + \varepsilon_k^2 \log^2 m_k^x\right)$$

$$\overset{(iii)}{\leq} \frac{(1 - \eta_k \varepsilon_k) D_\psi\left(\mu^{k,\star}, \mu^k\right) - D_\psi\left(\mu^{k,\star}, \mu^{k+1}\right)}{\eta_k} + \underline{\rho}_k + \underline{\sigma}_k$$

$$+ \eta_k X \cdot \underbrace{\frac{1}{X} \sum_{h,x_h,a_h} \frac{1}{p^x_{1:h}(x_h)} \left( \frac{\mathbb{I}^k_h \{x_h, a_h\}}{\mu^k_{1:h}(x_h, a_h) + \gamma_k} - 1 \right)}_{=: \underline{\tau}_k} + \eta_k X^2 A + \eta_k \varepsilon_k^2 X A \log^2 m_k^x$$

$$\leq \frac{(1 - \eta_k \varepsilon_k) D_\psi \left( \mu^{k,\star}, \mu^k \right) - D_\psi \left( \mu^{k,\star}, \mu^{k+1} \right)}{\eta_k} + \underline{\rho}_k + \underline{\sigma}_k + \eta_k X \underline{\tau}_k + \eta_k X^2 A + \eta_k \varepsilon_k^2 X A \log^2 m_k^x,$$

where $(i)$ is by Lemma D.1; recall $m_k^x := \min_{h,x_h,a_h} \left[ p^x \mu^k \right] (x_h, a_h)$ in $(ii)$; and $(iii)$ is due to Lemma B.1.

Rearranging shows that

$$D_\psi \left( \mu^{k+1,\star}, \mu^{k+1} \right)$$
$$\leq (1 - \eta_k \varepsilon_k) D_\psi \left( \mu^{k,\star}, \mu^k \right) + \eta_k \left( f_k \left( \mu^{k,\star}, \nu^k \right) - f_k \left( \mu^k, \nu^k \right) \right) + \eta_k^2 X A \varepsilon_k^2 \log^2 m_k^x$$
$$+ \eta_k^2 X \underline{\tau}_k + \eta_k^2 X^2 A + \eta_k \underline{\rho}_k + \eta_k \underline{\sigma}_k + \underbrace{D_\psi \left( \mu^{k+1,\star}, \mu^{k+1} \right) - D_\psi \left( \mu^{k,\star}, \mu^{k+1} \right)}_{=: \underline{\omega}_k}. \tag{30}$$

Analogously, for the min-player, we have

$$D_\psi \left( \nu^{k+1,\star}, \nu^{k+1} \right) \leq (1 - \eta_k \varepsilon_k) D_\psi \left( \nu^{k,\star}, \nu^k \right) + \eta_k \left( f_k \left( \mu^k, \nu^k \right) - f_k \left( \mu^k, \nu^{k,\star} \right) \right)$$
$$+ \eta_k^2 Y B \varepsilon_k^2 \log^2 m_k^y + \eta_k^2 Y \bar{\tau}_k + \eta_k^2 Y^2 B + \eta_k \bar{\rho}_k + \eta_k \bar{\sigma}_k + \bar{\omega}_k, \tag{31}$$

where

$$m_k^y := \min_{h,y_h,b_h} \left[ p^y \nu^k \right] (y_h, b_h),$$

$$\bar{\tau}_k := \frac{1}{Y} \sum_{h,y_h,b_h} \frac{1}{p^y_{1:h}(y_h)} \left( \frac{\mathbb{I}^k_h \{y_h, b_h\}}{\nu^k_{1:h}(y_h, b_h) + \gamma_k} - 1 \right),$$

$$\bar{\rho}_k := \sum_{h,y_h,b_h} \nu^k_{1:h}(y_h, b_h) \left[ \left( 1 - \left[ \boldsymbol{G}^\top \mu^k \right] (y_h, b_h) \right) - \frac{\mathbb{I}^k_h \{y_h, b_h\} r_h^k}{\nu^k_{1:h}(y_h, b_h) + \gamma_k} \right],$$

$$\bar{\sigma}_k := \sum_{h,y_h,b_h} \nu^{k,\star}_{1:h}(y_h, b_h) \left[ \frac{\mathbb{I}^k_h \{y_h, b_h\} r_h^k}{\nu^k_{1:h}(y_h, b_h) + \gamma_k} - \left( 1 - \left[ \boldsymbol{G}^\top \mu^k \right] (y_h, b_h) \right) \right],$$

$$\bar{\omega}_k := D_\psi \left( \nu^{k+1,\star}, \nu^{k+1} \right) - D_\psi \left( \nu^{k,\star}, \nu^{k+1} \right).$$

Combining Eq. (30) and Eq. (31), and noticing that $f_k \left( \mu^{k,\star}, \nu^k \right) - f_k \left( \mu^k, \nu^{k,\star} \right) \leq 0$, we have

$$D_\psi \left( \xi^{k+1,\star}, \xi^{k+1} \right)$$
$$\leq (1 - \eta_k \varepsilon_k) D_\psi \left( \xi^{k,\star}, \xi^k \right) + \eta_k^2 (X + Y) \tau_k + \eta_k^2 \left( X^2 A + Y^2 B \right) + \eta_k \rho_k + \eta_k \sigma_k + \omega_k$$
$$+ \eta_k^2 X A \varepsilon_k^2 \log^2 m_k^x + \eta_k^2 Y B \varepsilon_k^2 \log^2 m_k^y,$$

where $\tau_k := \underline{\tau}_k + \bar{\tau}_k$, $\rho_k := \underline{\rho}_k + \bar{\rho}_k$, $\sigma_k := \underline{\sigma}_k + \bar{\sigma}_k$, and $\omega_k := \underline{\omega}_k + \bar{\omega}_k$.

Now expanding the recursion in the above display leads to

$$D_\psi \left( \xi^{k+1,\star}, \xi^{k+1} \right)$$
$$\leq \underbrace{\sum_{i=1}^k w_k^i \eta_i \rho_i}_{\textbf{Term 1}} + \underbrace{\sum_{i=1}^k w_k^i \eta_i \sigma_i}_{\textbf{Term 2}} + \underbrace{X A (\log^2 m_k^x) \sum_{i=1}^k w_k^i (\eta_i \varepsilon_i)^2 + Y B (\log^2 m_k^y) \sum_{i=1}^k w_k^i (\eta_i \varepsilon_i)^2}_{\textbf{Term 3}}$$
$$+ \underbrace{\sum_{i=1}^k w_k^i \eta_i^2 (X + Y) \tau_i}_{\textbf{Term 4}} + \underbrace{\sum_{i=1}^k w_k^i \eta_i^2 \left( X^2 A + Y^2 B \right)}_{\textbf{Term 5}} + \underbrace{\sum_{i=1}^k w_k^i \omega_i}_{\textbf{Term 6}},$$

where $w_k^i = \prod_{j=i+1}^k (1 - \eta_j \varepsilon_j)$.

The proof is thus concluded. $\qquad\square$

C.2.1. BOUNDING CONTRACTION TERMS

**Lemma C.3** (Bounding **Term 1**). *When $k \geq k_0$, **Term 1** in Eq. (26) satisfies*

$$\textbf{Term 1} \leq (XA + YB)\ln(k)k^{-\alpha_\gamma + \alpha_\varepsilon} + k^{-\frac{\alpha_k}{2} + \frac{\alpha_\varepsilon}{2}} \log\left(k^2/\delta\right) .$$

*Proof.* Recall

$$\textbf{Term 1} = \sum_{i=1}^{k} w_k^i \eta_i \rho_i = \sum_{i=1}^{k} w_k^i \eta_i \underline{\rho}_i + \sum_{i=1}^{k} w_k^i \eta_i \bar{\rho}_i .$$

To bound $\sum_{i=1}^{k} w_k^i \eta_i \underline{\rho}_i$, note that

$$\sum_{i=1}^{k} w_k^i \eta_i \underline{\rho}_i$$

$$= \sum_{i=1}^{k} w_k^i \eta_i \left\langle \mu^i, \ell^i - \widehat{\ell^i} \right\rangle$$

$$\overset{(i)}{=} XA \sum_{i=1}^{k} w_k^i \eta_i \gamma_i + H \sqrt{2 \sum_{i=1}^{k} \left(w_k^i \eta_i\right)^2 \log \frac{k^2}{\delta}}$$

$$\lesssim XA \sum_{i=1}^{k} \left[ i^{-\alpha_\gamma - \alpha_\eta} \prod_{j=i+1}^{k} \left(1 - j^{-\alpha_\eta - \alpha_\varepsilon}\right) \right] + \sqrt{\log\left(\frac{k^2}{\delta}\right) \sum_{i=1}^{k} \left[ i^{-2\alpha_\eta} \left( \prod_{j=i+1}^{k} \left(1 - j^{-\alpha_\eta - \alpha_\varepsilon}\right) \right)^2 \right]}$$

$$\lesssim XA \sum_{i=1}^{k} \left[ i^{-\alpha_\gamma - \alpha_\eta} \prod_{j=i+1}^{k} \left(1 - j^{-\alpha_\eta - \alpha_\varepsilon}\right) \right] + \sqrt{\log\left(\frac{k^2}{\delta}\right) \sum_{i=1}^{k} \left[ i^{-2\alpha_\eta} \left( \prod_{j=i+1}^{k} \left(1 - j^{-\alpha_n - \alpha_\varepsilon}\right) \right) \right]}$$

$$\lesssim XA \ln(k) k^{-\alpha_\gamma + \alpha_\varepsilon} + \sqrt{\log\left(k^2/\delta\right) \ln(k) k^{-\alpha_\eta + \alpha_\varepsilon}}$$

$$\lesssim XA \ln(k) k^{-\alpha_\gamma + \alpha_\varepsilon} + k^{(\alpha_\varepsilon - \alpha_\eta)/2} \log\left(k^2/\delta\right) ,$$

where $(i)$ is given by Lemma G.1 and the last inequality comes from Lemma G.4 and the condition of $k \geq k_0$. Analogously, we have

$$\sum_{i=1}^{k} w_k^i \eta_i \bar{\rho}_i \lesssim YB \ln(k) k^{-\alpha_\gamma + \alpha_\varepsilon} + k^{(\alpha_\varepsilon - \alpha_\eta)/2} \log\left(k^2/\delta\right) .$$

The proof is completed by combining the upper bounds of $\sum_{i=1}^{k} w_k^i \eta_i \underline{\rho}_i$ and $\sum_{i=1}^{k} w_k^i \eta_i \bar{\rho}_i$. $\square$

**Lemma C.4** (Bounding **Term 2**). *When $k \geq k_0$, with probability $1 - (k^2/\delta)$, **Term 2** in Eq. (26) satisfies*

$$\textbf{Term 2} \leq k^{-\alpha_\eta + \alpha_\gamma} \log(k^2/\delta) .$$

*Proof.* Applying Lemma G.3 shows that with probability $1 - (k^2/\delta)$,

$$\textbf{Term 2} = \sum_{i=1}^{k} w_k^i \eta_i \sigma_i$$

$$= \sum_{i=1}^{k} w_k^i \eta_i \underline{\sigma}_i + \sum_{i=1}^{k} w_k^i \eta_i \bar{\sigma}_i$$

$$\lesssim \max_{1 \leq i \leq k} \frac{\eta_i w_k^i}{\gamma_k} \log \frac{k^2}{\delta}$$

$$\lesssim k^{-\alpha_\eta + \alpha_\gamma} \log \frac{k^2}{\delta} ,$$

where the last inequality is due to Lemma G.5 and the condition of $k \geq k_0$. □

**Lemma C.5** (Bounding **Term 3**). *When $k \geq k_0$, **Term 3** in Eq. (26) satisfies*

$$\textbf{Term 3} \lesssim \left( XA(\log^2 m_k^x) + YB(\log^2 m_k^y) \right) k^{-\alpha_\eta - \alpha_\varepsilon} .$$

*Proof.*

$$\begin{aligned}
\textbf{Term 3} =& XA(\log^2 m_k^x) \sum_{i=1}^{k} w_k^i \left( \eta_i \varepsilon_i \right)^2 + YB(\log^2 m_k^y) \sum_{i=1}^{k} w_k^i \left( \eta_i \varepsilon_i \right)^2 \\
\lesssim& \left( XA(\log^2 m_k^x) + YB(\log^2 m_k^y) \right) k^{-2(\alpha_\eta + \alpha_\varepsilon) + \alpha_\eta + \alpha_\varepsilon} \\
=& \left( XA(\log^2 m_k^x) + YB(\log^2 m_k^y) \right) k^{-\alpha_\eta - \alpha_\varepsilon} ,
\end{aligned}$$

where the inequality follows from Lemma G.4 and the condition of $k \geq k_0$. □

**Lemma C.6** (Bounding **Term 4**). *With probability $1 - \delta$, **Term 4** in Eq. (26) satisfies*

$$\textbf{Term 4} \leq k^{\alpha_\gamma - 2\alpha_\eta}(X + Y) \log \left( 1/\delta \right) .$$

*Proof.*

**Term 4**

$$= \sum_{i=1}^{k} w_k^i \eta_i^2 \left( X \underline{\tau}_i + Y \bar{\tau}_i \right)$$

$$= \sum_{i=1}^{k} w_k^i \eta_i^2 \left( X \cdot \frac{1}{X} \sum_{h,x_h,a_h} \frac{1}{p_{1:h}^x(x_h)} \left( \frac{\mathbb{I}_h^k \{x_h, a_h\}}{\mu_{1:h}^k(x_h, a_h) + \gamma_k} - 1 \right) + Y \cdot \frac{1}{Y} \sum_{h,y_h,b_h} \frac{1}{p_{1:h}^y(y_h)} \left( \frac{\mathbb{I}_h^k \{y_h, b_h\}}{\nu_{1:h}^k(y_h, b_h) + \gamma_k} - 1 \right) \right)$$

$$\leq \max_{1 \leq i \leq k} \frac{w_k^i \eta_i^2 (X + Y)}{\gamma_k} \log(1/\delta)$$

$$\leq k^{\alpha_\gamma - 2\alpha_\eta}(X + Y) \log \left( 1/\delta \right) ,$$

where the first inequality follows from that $1/X \leq p_{1:h}^x(x_h)$ for all $(x_h, a_h)$ guaranteed by Lemma B.1 and Lemma G.3. □

**Lemma C.7** (Bounding **Term 5**). *When $k \geq k_0$, **Term 5** in Eq. (26) satisfies*

$$\textbf{Term 5} \lesssim \left( X^2 A + Y^2 B \right) k^{-\alpha_\eta + \alpha_\varepsilon} .$$

*Proof.*

$$\begin{aligned}
\textbf{Term 5} =& \sum_{i=1}^{k} w_k^i \eta_i^2 \left( X^2 A + Y^2 B \right) \\
\lesssim& \left( X^2 A + Y^2 B \right) k^{-2\alpha_\eta + \alpha_\eta + \alpha_\varepsilon} \\
=& \left( X^2 A + Y^2 B \right) k^{-\alpha_\eta + \alpha_\varepsilon} ,
\end{aligned}$$

where the inequality is due to Lemma G.4 as well as the condition of $k \geq k_0$. □

**Lemma C.8** (Bounding **Term 6**). *When $k \geq k_0$, **Term 6** in Eq. (26) satisfies*

$$\textbf{Term 6} \lesssim (X+Y)^{\frac{1}{2}} \log^2\left(1/\left(m_k^x m_k^y\right)\right) \log(k) k^{-\min\{1,(3-\alpha_\varepsilon)/2\}+\alpha_\eta+\alpha_\epsilon} .$$

*Proof.* By definition, we have

$$\textbf{Term 6} = \sum_{i=1}^{k} w_k^i \omega_i$$

$$\lesssim (X+Y)^{\frac{1}{2}} \log^2\left(1/\left(m_k^x m_k^y\right)\right) \sum_{i=1}^{k} w_k^i i^{-\min\{1,(3-\alpha_\varepsilon)/2\}}$$

$$\lesssim (X+Y)^{\frac{1}{2}} \log^2\left(1/\left(m_k^x m_k^y\right)\right) \log(k) k^{-\min\{1,(3-\alpha_\varepsilon)/2\}+\alpha_\eta+\alpha_\epsilon} ,$$

where the first inequality is due to Lemma D.2 and the second inequality comes from Lemma G.4.

$\square$

# D. Omitted Details in the Proof of Theorem 5.1

## D.1. One-step Analysis of OMD with Virtual Transition-Weighted Negentropy Regularized Loss

**Lemma D.1.** *Let*

$$\mu' = \operatorname*{arg\,min}_{\tilde{\mu} \in \Omega} \sum_{h,x_h,a_h} \tilde{\mu}_{1:h}\left(x_h, a_h\right)\left(\ell\left(x_h, a_h\right) + \varepsilon p_{1:h}^x\left(x_h\right) \log\left[p^x \mu\right]\left(x_h, a_h\right)\right) + \frac{1}{\eta} D_\psi(\tilde{\mu}, \mu) ,$$

*for some convex set $\Omega \subseteq \Pi_{\max}, \ell \in \mathbb{R}_{\geq 0}^{XA}$, and $\varepsilon \in \left[0, \frac{1}{\eta}\right]^{XA}$. Then $\forall u \in \Omega$.*

$$\langle \mu - u, \ell + \varepsilon p^x \log\left[p^x \mu\right] \rangle$$
$$\leq \frac{1}{\eta}\left(D_\psi(u, \mu) - D_\psi(u, \mu')\right) + \sum_{h,x_h,a_h}\left(\frac{\eta}{p_{1:h}^x(x_h)}\mu_{1:h}(x_h, a_h)\ell(x_h, a_h)^2 + \eta\varepsilon^2 \log^2\left[p^x \mu\right](x_h, a_h)\right) ,$$

*where $\left[p^x \log\left[p^x \mu\right]\right]\left(x_h, a_h\right) := p_{1:h}^x(x_h) \log\left[p^x \mu\right]\left(x_h, a_h\right)$.*

*Proof.* The common one-step analysis of OMD shows that

$$\langle \mu' - u, \ell + \varepsilon p^x \log\left[p^x \mu\right] \rangle \leq \frac{1}{\eta}\left(D_\psi(u, \mu) - D_\psi(u, \mu') - D_\psi(\mu', \mu)\right) .$$

Hence

$$\langle \mu - u, \ell + \varepsilon p^x \log\left[p^x \mu\right] \rangle$$
$$\leq \frac{1}{\eta}\left(D_\psi(u, \mu) - D_\psi(u, \mu') - D_\psi(\mu', \mu)\right) + \langle \mu - \mu', \ell + \varepsilon p^x \log\left[p^x \mu\right] \rangle . \tag{32}$$

Then, to upper bound $\langle \mu - \mu', \ell + \varepsilon p^x \log\left[p^x \mu\right] \rangle - \frac{1}{\eta} D_\psi\left(\mu', \mu\right)$, notice that

$$\langle \mu - \mu', \ell + \varepsilon p^x \log\left[p^x \mu\right] \rangle - \frac{1}{\eta} D_\psi\left(\mu', \mu\right)$$
$$\leq \sup_{\nu \in \mathbb{R}_+^{XA}}\left(\langle \mu - \nu, \ell + \varepsilon p^x \log\left[p^x \mu\right] \rangle - \frac{1}{\eta} D_\psi\left(\nu, \mu\right)\right)$$
$$= \langle \mu, \ell + \varepsilon p^x \log\left[p^x \mu\right] \rangle - \inf_{\nu \in \mathbb{R}_+^{XA}}\left(\langle \nu, \ell + \varepsilon p^x \log\left[p^x \mu\right] \rangle + \frac{1}{\eta} D_\psi\left(\nu, \mu\right)\right) . \tag{33}$$

Further, the first-order optimality condition, $\ell + \varepsilon p^x \log [p^x \mu] + \frac{1}{\eta}(\nabla \psi(\nu) - \nabla \psi(\mu)) = 0$, implies that

$$\log \frac{\nu_{1:h}(x_h, a_h)}{\mu_{1:h}(x_h, a_h)} = -\frac{\eta}{p^x_{1:h}(x_h)} \left[ \ell(x_h, a_h) + \varepsilon p^x_{1:h}(x_h) \log [p^x \mu](x_h, a_h) \right] ,$$

and thus

$$
\begin{aligned}
&\nu_{1:h}(x_h, a_h) \\
&= \mu_{1:h}(x_h, a_h) \exp \left( -\frac{\eta}{p^x_{1:h}(x_h)} \left[ \ell(x_h, a_h) + \varepsilon p^x_{1:h}(x_h) \log [p^x \mu](x_h, a_h) \right] \right) .
\end{aligned}
\tag{34}
$$

Substituting Eq. (34) into Eq. (33) leads to

$$
\begin{aligned}
& \langle \mu - \mu', \ell + \varepsilon p^x \log [p^x \mu] \rangle - \frac{1}{\eta} D_\psi(\mu', \mu) \\
=& \sum_{h, x_h, a_h} \left[ (\mu_{1:h}(x_h, a_h) - \nu_{1:h}(x_h, a_h)) (\ell(x_h, a_h) + \varepsilon p^x_{1:h}(x_h) \log [p^x \mu](x_h, a_h)) \right. \\
& \left. - \frac{1}{\eta} \left( [p^x \nu](x_h, a_h) \log \frac{\nu_{1:h}(x_h, a_h)}{\mu_{1:h}(x_h, a_h)} - ([p^x \nu](x_h, a_h) - [p^x \mu](x_h, a_h)) \right) \right] \\
=& \sum_{h, x_h, a_h} \left[ \mu_{1:h}(x_h, a_h) (\ell(x_h, a_h) + \varepsilon p^x_{1:h}(x_h) \log [p^x \mu](x_h, a_h)) \right. \\
& \left. + \frac{[p^x \mu](x_h, a_h)}{\eta} \left( \exp \left( -\frac{\eta}{p^x_{1:h}(x_h)} \left[ \ell(x_h, a_h) + \varepsilon p^x_{1:h}(x_h) \log [p^x \mu](x_h, a_h) \right] \right) - 1 \right) \right] \\
=& \sum_{h, x_h, a_h} \frac{[p^x \mu](x_h, a_h)}{\eta} \left[ \frac{\eta}{p^x_{1:h}(x_h)} (\ell(x_h, a_h) + \varepsilon p^x_{1:h}(x_h) \log [p^x \mu](x_h, a_h)) \right. \\
& \left. + \exp \left( -\frac{\eta}{p^x_{1:h}(x_h)} \left[ \ell(x_h, a_h) + \varepsilon p^x_{1:h}(x_h) [p^x \mu](x_h, a_h) \right] \right) - 1 \right] \\
\overset{(i)}{\leq}& \sum_{h, x_h, a_h} \frac{\eta}{p^x_{1:h}(x_h)} \mu_{1:h}(x_h, a_h) \ell^2(x_h, a_h) \\
& + \sum_{h, x_h, a_h} \frac{p^x_{1:h}(x_h)}{\eta} \left[ \mu_{1:h}(x_h, a_h) \frac{\eta}{p^x_{1:h}(x_h)} \varepsilon p^x_{1:h}(x_h) \log [p^x \mu](x_h, a_h) \right. \\
& + \mu_{1:h}(x_h, a_h) \exp \left( -\frac{\eta}{p^x_{1:h}(x_h)} \left[ \ell(x_h, a_h) + \varepsilon p^x_{1:h}(x_h) \log [p^x \mu](x_h, a_h) \right] \right) \\
& \left. - \mu_{1:h}(x_h, a_h) \exp \left( -\frac{\eta}{p^x_{1:h}(x_h)} \ell(x_h, a_h) \right) \right] \\
=& \sum_{h, x_h, a_h} \frac{\eta}{p^x_{1:h}(x_h)} \mu_{1:h}(x_h, a_h) \ell^2(x_h, a_h) \\
& + \sum_{h, x_h, a_h} \frac{1}{\eta} \left[ \eta \varepsilon [p^x \mu](x_h, a_h) \log [p^x \mu](x_h, a_h) \right. \\
& \left. + \exp \left( -\frac{\eta}{p^x_{1:h}(x_h)} \ell(x_h, a_h) \right) ([p^x \mu](x_h, a_h)^{1-\eta\varepsilon} - [p^x \mu](x_h, a_h)) \right] \\
\overset{(ii)}{\leq}& \sum_{h, x_h, a_h} \frac{\eta}{p^x_{1:h}(x_h)} \mu_{1:h}(x_h, a_h) \ell^2(x_h, a_h) \\
& + \sum_{h, x_h, a_h} \frac{1}{\eta} \left[ \eta \varepsilon [p^x \mu](x_h, a_h) \log [p^x \mu](x_h, a_h) + [p^x \mu](x_h, a_h)^{1-\eta\varepsilon} - [p^x \mu](x_h, a_h) \right] \\
\overset{(iii)}{\leq}& \sum_{h, x_h, a_h} \frac{\eta}{p^x_{1:h}(x_h)} \mu_{1:h}(x_h, a_h) \ell^2(x_h, a_h)
\end{aligned}
$$

$$+ \sum_{h,x_h,a_h} \frac{1}{\eta} \left[ \eta \varepsilon \left[ p^x \mu \right] (x_h, a_h) \log \left[ p^x \mu \right] (x_h, a_h) - \eta \varepsilon \left[ p^x \mu \right] (x_h, a_h)^{1-\eta\varepsilon} \log \left[ p^x \mu \right] (x_h, a_h) \right]$$

$$= \sum_{h,x_h,a_h} \frac{\eta}{p^x_{1:h}(x_h)} \mu_{1:h}(x_h, a_h) \ell^2(x_h, a_h)$$

$$- \sum_{h,x_h,a_h} \frac{1}{\eta} \left( \eta \varepsilon \left( \left[ p^x \mu \right] (x_h, a_h)^{1-\eta\varepsilon} - \left[ p^x \mu \right] (x_h, a_h) \right) \log \left[ p^x \mu \right] (x_h, a_h) \right)$$

$$\overset{(iv)}{\leq} \sum_{h,x_h,a_h} \frac{\eta}{p^x_{1:h}(x_h)} \mu_{1:h}(x_h, a_h) \ell^2(x_h, a_h)$$

$$+ \sum_{h,x_h,a_h} \frac{1}{\eta} \left[ \eta^2 \varepsilon^2 \left[ p^x \mu \right] (x_h, a_h)^{1-\eta\varepsilon} \log^2 \left[ p^x \mu \right] (x_h, a_h) \right]$$

$$\leq \sum_{h,x_h,a_h} \frac{\eta}{p^x_{1:h}(x_h)} \mu_{1:h}(x_h, a_h) \ell^2(x_h, a_h) + \sum_{h,x_h,a_h} \eta \varepsilon^2 \log^2 \left[ p^x \mu \right] (x_h, a_h), \tag{35}$$

where in $(i)$ we use $e^{-x} \leq x^2 - x + 1$ for all $x \geq 0$ and $\frac{\eta}{p^x_{1:h}(x_h)} \ell(x_h, a_h) \geq 0$; $(ii)$ follows from $\frac{\eta}{p^x_{1:h}} \ell(x_h, a_h) \geq 0$ and $\left( \left[ p^x \mu \right] (x_h, a_h) \right)^{1-\eta\varepsilon} - \left[ p^x \mu \right] (x_h, a_h) \geq 0$; $(iii)$ and $(iv)$ are by Lemma G.6.

Substituting Eq. (35) into Eq. (32) finishes the proof. $\qquad \square$

### D.2. Bounding Divergence Differences

Recall the Bregman divergence difference $\omega_k := \underline{\omega}_k + \bar{\omega}_k$, where $\underline{\omega}_k = D_\psi \left( \mu^{k+1,\star}, \mu^{k+1} \right) - D_\psi \left( \mu^{k,\star}, \mu^{k+1} \right)$ and $\bar{\omega}_k = D_\psi \left( \nu^{k+1,\star}, \nu^{k+1} \right) - D_\psi \left( \nu^{k,\star}, \nu^{k+1} \right)$.

**Lemma D.2.** *The Bregman divergence $\omega_k$ can be bounded as*

$$|\omega_k| = \mathcal{O} \left( \frac{(X+Y)^{\frac{1}{2}} \log^2 \left( 1/ \left( m^x_k m^y_k \right) \right)}{k^{\min\{1,(3-\alpha_\varepsilon)/2\}}} \right).$$

*Proof.* By definition, we have

$$|\omega_k| \leq \left| D_\psi \left( \mu^{k+1,\star}, \mu^{k+1} \right) - D_\psi \left( \mu^{k,\star}, \mu^{k+1} \right) \right| + \left| D_\psi \left( \nu^{k+1,\star}, \nu^{k+1} \right) - D_\psi \left( \nu^{k,\star}, \nu^{k+1} \right) \right|$$

$$\lesssim \left\| p^x \mu^{k+1,\star} - p^x \mu^{k,\star} \right\|_1 \log \frac{1}{m^x_k} + \left\| p^y \nu^{k+1,\star} - p^y \nu^{k,\star} \right\|_1 \log \frac{1}{m^y_k}$$

$$\lesssim \frac{(X+Y)^{\frac{1}{2}} \log^2 \left( 1/ \left( m^x_k m^y_k \right) \right)}{k^{\min\{1,(3-\alpha_\varepsilon)/2\}}},$$

where the second inequality is due to Lemma D.3 and the last inequality comes from Lemma D.4. $\qquad \square$

**Lemma D.3** (Bounding divergence using $\ell_1$-norm)**.** *For all $\mu, \mu^1, \mu^2 \in \Pi^k_{\max}$, it holds that*

$$\left| D_\psi \left( \mu^1, \mu \right) - D_\psi \left( \mu^2, \mu \right) \right| \leq \mathcal{O} \left( \left\| p^x \mu^1 - p^x \mu^2 \right\|_1 \log \frac{1}{m^x_k} \right).$$

*Proof.* By definition of virtual transition-weighted negentropy $\psi$, one can deduce that

$$D_\psi \left( \mu^1, \mu \right) - D_\psi \left( \mu^2, \mu \right)$$

$$= \sum_{h,x_h,a_h} p^x_{1:h}(x_h) \left( \mu^1_{1:h}(x_h, a_h) \log \frac{\mu^1_{1:h}(x_h, a_h)}{\mu_{1:h}(x_h, a_h)} - \mu^2_{1:h}(x_h, a_h) \log \frac{\mu^2_{1:h}(x_h, a_h)}{\mu_{1:h}(x_h, a_h)} \right)$$

$$= \sum_{h,x_h,a_h} p^x_{1:h}(x_h) \left( \mu^1_{1:h}(x_h, a_h) - \mu^2_{1:h}(x_h, a_h) \right) \log \frac{\mu^1_{1:h}(x_h, a_h)}{\mu_{1:h}(x_h, a_h)}$$

$$+ \sum_{h, x_h, a_h} p_{1:h}^x (x_h) \, \mu_{1:h}^2 (x_h, a_h) \left( \log \frac{\mu_{1:h}^1 (x_h, a_h)}{\mu_{1:h} (x_h, a_h)} - \log \frac{\mu_{1:h}^2 (x_h, a_h)}{\mu_{1:h} (x_h, a_h)} \right)$$

$$\leq \mathcal{O} \left( \left\| p^x \mu^1 - p^x \mu^2 \right\|_1 \log \frac{1}{m_k^x} \right) - D_\psi (\mu^2, \mu^1)$$

$$\leq \mathcal{O} \left( \left\| p^x \mu^1 - p^x \mu^2 \right\|_1 \log \frac{1}{m_k^x} \right) ,$$

where the first inequality is due to that $p_{1:h}^x (x_h) \leq 1$ for all $x_h \in \mathcal{X}$ and thus $\min_{h, x_h} \mu_{1:h}(x_h) \geq m_k^x$. $\qquad \square$

**Lemma D.4.** *Let $p^z \xi^{k,\star} = (p^x \mu^{k,\star}, p^y \nu^{k,\star})$. The $\ell_1$-norm of the virtual transition-weighted difference between $\xi^{k,\star}$ and $\xi^{k+1,\star}$ satisfies*

$$\left\| p^z \xi^{k+1,\star} - p^z \xi^{k,\star} \right\|_1 = \mathcal{O} \left( \frac{(X + Y)^{\frac{1}{2}} \log \left( 1/(m_k^x m_k^y) \right)}{k^{\min\{1, (3 - \alpha_\varepsilon)/2\}}} \right) .$$

*Proof.* Recall the entropy perturbed game $f_k(\mu, \nu)$ defined in Eq. (4). To begin with, note that for all $k \geq 1$ and $(\mu, \nu) \in \Pi_{\max}^k \times \Pi_{\min}^k$, it holds that

$$
\begin{aligned}
f_k \left( \mu, \nu^{k,\star} \right) - f_k \left( \mu^{k,\star}, \nu \right) = & f_k \left( \mu, \nu^{k,\star} \right) - f_k \left( \mu^{k,\star}, \nu^{k,\star} \right) + f_k \left( \mu^{k,\star}, \nu^{k,\star} \right) - f_k \left( \mu^{k,\star}, \nu \right) \\
& \overset{(i)}{\geqslant} f_k \left( \mu, \nu^{k,\star} \right) - f_k \left( \mu^{k,\star}, \nu^{k,\star} \right) - \nabla_\mu f_k \left( \mu^{k,\star}, \nu^{k,\star} \right)^\top \left( \mu - \mu^{k,\star} \right) \\
& \quad - f_k \left( \mu^{k,\star}, \nu \right) - \left( -f_k \left( \mu^{k,\star}, \nu^{k,\star} \right) \right) - \left( -\nabla_\nu f_k \left( \mu^{k,\star}, \nu^{k,\star} \right)^\top \left( \nu - \nu^{k,\star} \right) \right) \\
& \geqslant \varepsilon_k D_\psi \left( \mu, \mu^{k,\star} \right) + \varepsilon_k D_\psi \left( \nu, \nu^{k,\star} \right) \\
& = \varepsilon_k \, \mathrm{KL} \left( p^x \mu, p^x \mu^{k,\star} \right) + \varepsilon_k \, \mathrm{KL} \left( p^y \nu, p^y \nu^{k,\star} \right) \\
& \overset{(ii)}{\geqslant} \frac{1}{2} \varepsilon_k \left( \left\| p^x \mu - p^x \mu^{k,\star} \right\|_1^2 + \left\| p^y \nu - p^y \nu^{k,\star} \right\|_1^2 \right) \\
& \geqslant \frac{1}{4} \varepsilon_k \left\| p^z \xi - p^z \xi^{k,\star} \right\|_1^2 ,
\end{aligned}
\tag{36}
$$

where $(i)$ follows from the fact that $\mu^{k,\star}$ is the minimizer of $f_k$ given $\nu^{k,\star}$ and $\nu^{k,\star}$ is the maximizer of $f_k$ given $\mu^{k,\star}$ together with the first-order optimality condition; and $(ii)$ is by Pinsker's inequality.

Let $p_k = \min\{1, 2k^{-3}\}$ and $\bar{\mu}$ and $\bar{\nu}$ are the uniform policies for the max-player and the min-player, respectively. Define $\mu^{k+1,'} = p_{k+1} \bar{\mu} + (1 - p_{k+1}) \mu^{k+1,\star}$. Then for all $h$ and $(x_h, a_h)$, it is clear that

$$\mu^{k+1,'} (a_h | x_h) \geqslant p_{k+1} \frac{1}{A} + (1 - p_{k+1}) \frac{1}{A(k+1)^2} \geqslant \frac{1}{Ak^2} ,$$

which means that $\mu^{k+1,'} \in \Pi_{\max}^k$. Similarly, we define $\nu^{k+1,'}$, which satisfies that $\nu^{k+1,'} \in \Pi_{\min}^k$. Thus, by Eq. (36), we have

$$f_k \left( \mu^{k+1,'}, \nu^{k,\star} \right) - f_k \left( \mu^{k,\star}, \nu^{k+1,'} \right) \geqslant \frac{1}{4} \varepsilon_k \left\| p^z \xi^{k+1,'} - p^z \xi^{k,\star} \right\|_1^2 . \tag{37}$$

On the other hand, since $\left( \mu^{k,\star}, \nu^{k,\star} \right) \in \Pi_{\max}^{k+1} \times \Pi_{\min}^{k+1}$, we have

$$f_{k+1} \left( \mu^{k,\star}, \nu^{k+1,\star} \right) - f_{k+1} \left( \mu^{k+1,\star}, \nu^{k,\star} \right) \geqslant \frac{1}{4} \varepsilon_{k+1} \left\| p^z \xi^{k,\star} - p^z \xi^{k+1,\star} \right\|_1^2 . \tag{38}$$

Therefore, one can see that

$$
\begin{aligned}
& f_k \left( \mu^{k+1,\star}, \nu^{k,\star} \right) - f_k \left( \mu^{k,\star}, \nu^{k+1,\star} \right) \\
= & f_k \left( \mu^{k+1,'}, \nu^{k,\star} \right) - f_k \left( \mu^{k,\star}, \nu^{k+1,'} \right) + f_k \left( \mu^{k+1,\star}, \nu^{k,\star} \right) - f_k \left( \mu^{k+1,'}, \nu^{k,\star} \right) + f_k \left( \mu^{k,\star}, \nu^{k+1,'} \right) - f_k \left( \mu^{k,\star}, \nu^{k+1,\star} \right) \\
\geq & f_k \left( \mu^{k+1,'}, \nu^{k,\star} \right) - f_k \left( \mu^{k,\star}, \nu^{k+1,'} \right) + \left\langle \nabla_\mu f_k \left( \mu^{k+1,'}, \nu^{k,\star} \right), \mu^{k+1,\star} - \mu^{k+1,'} \right\rangle
\end{aligned}
$$

$$+ \left\langle \nabla_\nu f_k \left( \mu^{k,\star}, \nu^{k+1,\prime} \right), \nu^{k+1,\prime} - \nu^{k+1,\star} \right\rangle$$

$$\geq \frac{1}{4} \varepsilon_k \left\| p^z \xi^{k+1,\prime} - p^z \xi^{k,\star} \right\|_1^2 - \left\| \nabla_\mu f_k \left( \mu^{k+1,\prime}, \nu^{k,\star} \right) \right\|_\infty \left\| \mu^{k+1,\star} - \mu^{k+1,\prime} \right\|_1$$

$$- \left\| \nabla_\nu f_k \left( \mu^{k,\star}, \nu^{k+1,\prime} \right) \right\|_\infty \left\| \nu^{k+1,\prime} - \nu^{k+1,\star} \right\|_1$$

$$\geq \frac{1}{4} \varepsilon_k \left\| p^z \xi^{k+1,\prime} - p^z \xi^{k,\star} \right\|_1^2 - \sup_{\mu \in \Pi_{\max}^{k+1}} \left\| \nabla_\mu f_k \left( \mu, \nu^{k,\star} \right) \right\|_\infty \left\| \mu^{k+1,\star} - \mu^{k+1,\prime} \right\|_1$$

$$- \sup_{\nu \in \Pi_{\min}^{k+1}} \left\| \nabla_\nu f_k (\mu^{k,\star}, \nu) \right\|_\infty \left\| \nu^{k+1,\prime} - \nu^{k+1,\star} \right\|_1, \tag{39}$$

where the first inequality is due to that $f_k$ is convex in $\mu$ and is concave in $\nu$ and the second inequality follows from Eq. (37) and Höder's inequality. Further, by noticing that

$$\sup_{\mu \in \Pi_{\max}^{k+1}} \left\| \nabla_\mu f_k \left( \mu, \nu^{k,\star} \right) \right\|_\infty = \sup_{\mu \in \Pi_{\max}^{k+1}} \max_{h, x_h, a_h} \left| \left[ \boldsymbol{G} \nu^{k,\star} \right] (x_h, a_h) + \varepsilon_k p_{1:h}^x (x_h) \log \left[ p^x \mu \right] (x_h, a_h) \right|$$

$$\leq \sup_{\mu \in \Pi_{\max}^{k+1}} \max_{h, x_h, a_h} \left| \left[ \boldsymbol{G} \nu^{k,\star} \right] (x_h, a_h) \right| + \left| \varepsilon_k p_{1:h}^x (x_h) \log \left[ p^x \mu \right] (x_h, a_h) \right|$$

$$\leq 1 + k^{-\alpha_\varepsilon} \log \frac{1}{m_{k+1}^x} = \mathcal{O}(1),$$

and

$$\left\| \mu^{k+1,\star} - \mu^{k+1,\prime} \right\|_1 = \left\| p_{k+1} \left( \bar\mu - \mu_{k+1}^\star \right) \right\|_1 \leq \left\| p_{k+1} \bar\mu \right\|_1 + \left\| p_{k+1} \mu_{k+1}^\star \right\|_1 = \mathcal{O} \left( X / k^3 \right),$$

we proceed to lower bound Eq. (39) as

$$f_k \left( \mu^{k+1,\star}, \nu^{k,\star} \right) - f_k \left( \mu^{k,\star}, \nu^{k+1,\star} \right)$$

$$\geq \frac{1}{8} \varepsilon_k \left\| p^z \xi^{k+1,\star} - p^z \xi^{k,\star} \right\|_1^2 - \frac{1}{4} \varepsilon_k \left\| p^z \xi^{k+1,\prime} - p^z \xi^{k+1,\star} \right\|_1^2 - \mathcal{O} \left( \frac{X + Y}{k^3} \right)$$

$$\geq \frac{1}{8} \varepsilon_k \left\| p^z \xi^{k+1,\star} - p^z \xi^{k,\star} \right\|_1^2 - \frac{2}{4} \varepsilon_k \left( \left\| p^x \mu^{k+1,\prime} - p^x \mu^{k+1,\star} \right\|_1^2 + \left\| p^y \nu^{k+1,\prime} - p^y \nu^{k+1,\star} \right\|_1^2 \right)$$

$$- \mathcal{O} \left( \frac{X + Y}{k^3} \right)$$

$$\geq \frac{1}{8} \varepsilon_k \left\| p^z \xi^{k+1,\star} - p^z \xi^{k,\star} \right\|_1^2 - \mathcal{O} \left( \frac{X + Y}{k^3} \right)$$

$$- \varepsilon_k \left( \left\| p_{k+1} \cdot p^x \bar\mu \right\|_1^2 + \left\| p_{k+1} \cdot p^x \mu^{k+1,\star} \right\|_1^2 + \left\| p_{k+1} \cdot p^y \bar\nu \right\|_1^2 + \left\| p_{k+1} \cdot p^y \nu^{k+1,\star} \right\|_1^2 \right)$$

$$= \frac{1}{8} \varepsilon_k \left\| p^z \xi^{k+1,\star} - p^z \xi^{k,\star} \right\|_1^2 - \mathcal{O} \left( \frac{X + Y}{k^3} \right) - \mathcal{O} \left( \frac{1}{k^{6 + \alpha_\varepsilon}} \right)$$

$$\geq \frac{1}{8} \varepsilon_{k+1} \left\| p^z \xi^{k+1,\star} - p^z \xi^{k,\star} \right\|_1^2 - \mathcal{O} \left( \frac{X + Y}{k^3} \right), \tag{40}$$

where the first and the second inequality is by $a^2 + b^2 \geq (a + b)^2 / 2$ for any $a$ and $b$.

Combining Eq. (40) with Eq. (38) shows that

$$\frac{3}{8} \varepsilon_{k+1} \| p^z \xi^{k+1,\star} - p^z \xi^{k,\star} \|_1^2$$

$$\leq f_{k+1} \left( \mu^{k,\star}, \nu^{k+1,\star} \right) - f_k \left( \mu^{k,\star}, \nu^{k+1,\star} \right) - f_{k+1} \left( \mu^{k+1,\star}, \nu^{k,\star} \right) + f_k \left( \mu^{k+1,\star}, \nu^{k,\star} \right) + \mathcal{O} \left( \frac{X + Y}{k^3} \right)$$

$$\overset{(i)}{=} \bar f_k \left( \mu^{k,\star}, \nu^{k+1,\star} \right) - \bar f_k \left( \mu^{k+1,\star}, \nu^{k,\star} \right) + \mathcal{O} \left( \frac{X + Y}{k^3} \right)$$

$$= \bar f_k \left( \mu^{k,\star}, \nu^{k+1,\star} \right) - \bar f_k \left( \mu^{k+1,\star}, \nu^{k+1,\star} \right) + \bar f_k \left( \mu^{k+1,\star}, \nu^{k+1,\star} \right) - \bar f_k \left( \mu^{k+1,\star}, \nu^{k,\star} \right) + \mathcal{O} \left( \frac{X + Y}{k^3} \right)$$

$$\leq \left\langle \nabla_\mu \bar{f}_k \left( \mu^{k,\star}, \nu^{k+1,\star} \right), \mu^{k,\star} - \mu^{k+1,\star} \right\rangle + \left\langle \nabla_\nu \bar{f}_k \left( \mu^{k+1,\star}, \nu^{k,\star} \right), \nu^{k+1,\star} - \nu^{k,\star} \right\rangle + \mathcal{O} \left( \frac{X+Y}{k^3} \right)$$

$$= \left\langle \nabla_\mu \bar{f}_k \left( \mu^{k,\star}, \nu^{k+1,\star} \right) / p^x, p^x \left( \mu^{k,\star} - \mu^{k+1,\star} \right) \right\rangle + \left\langle \nabla_\nu \bar{f}_k \left( \mu^{k+1,\star}, \nu^{k,\star} \right) / p^y, p^y \left( \nu^{k+1,\star} - \nu^{k,\star} \right) \right\rangle + \mathcal{O} \left( \frac{X+Y}{k^3} \right)$$

$$\leq \left\| \nabla_\mu \bar{f}_k \left( \mu^{k,\star}, \nu^{k+1,\star} \right) / p^x \right\|_\infty \| p^x \left( \mu^{k,\star} - \mu^{k+1,\star} \right) \|_1 + \left\| \nabla_\nu \bar{f}_k \left( \mu^{k+1,\star}, \nu^{k,\star} \right) / p^y \right\|_\infty \left\| p^y \left( \nu^{k+1,\star} - \nu^{k,\star} \right) \right\|_1$$
$$+ \mathcal{O} \left( \frac{X+Y}{k^3} \right)$$

$$\leq \left( \sup_{\mu \in \Pi_{\max}^k} \left\| \nabla_\mu \bar{f}_k \left( \mu, \nu^{k+1,\star} \right) / p^x \right\|_\infty + \sup_{\nu \in \Pi_{\min}^k} \left\| \nabla_\nu \bar{f}_k \left( \mu^{k+1,\star}, \nu \right) / p^y \right\|_\infty \right) \left\| p^z \xi^{k+1,\star} - p^z \xi^{k,\star} \right\|_1 + \mathcal{O} \left( \frac{X+Y}{k^3} \right)$$

$$\leq \left( \sup_{\mu \in \Pi_{\max}^k} \max_{h,x_h,a_h} \left| (\varepsilon_k - \varepsilon_{k+1}) \log \left[ p^x \mu \right] (x_h, a_h) \right| + \sup_{\nu \in \Pi_{\min}^k} \max_{h,y_h,b_h} \left| (\varepsilon_k - \varepsilon_{k+1}) \log \left[ p^y \nu \right] (y_h, b_h) \right| \right)$$
$$\cdot \left\| p^z \xi^{k+1,\star} - p^z \xi^{k,\star} \right\|_1 + \mathcal{O} \left( \frac{X+Y}{k^3} \right)$$

$$= \mathcal{O} \left( (\varepsilon_k - \varepsilon_{k+1}) \left( \log \frac{1}{m_k^x} + \log \frac{1}{m_k^y} \right) \left\| p^z \xi^{k+1,\star} - p^z \xi^{k,\star} \right\|_1 + \frac{X+Y}{k^3} \right), \tag{41}$$

where in $(i)$ we let $\bar{f}_k(\mu, \nu) \coloneqq f_{k+1}(\mu, \nu) - f_k(\mu, \nu)$.

Solving the quadratic equation in Eq. (41) leads to

$$\left\| p^z \xi^{k+1,\star} - p^z \xi^{k,\star} \right\|_1 \lesssim \frac{(\varepsilon_k - \varepsilon_{k+1}) \log \left( 1/(m_k^x m_k^y) \right) + \sqrt{(\varepsilon_k - \varepsilon_{k+1})^2 \log^2 \left( m_k^x m_k^y \right) + \varepsilon_{k+1} (X+Y)/k^3}}{\varepsilon_{k+1}}$$

$$\lesssim \frac{(\varepsilon_k - \varepsilon_{k+1})}{\varepsilon_{k+1}} \log \left( 1/(m_k^x m_k^y) \right) + \sqrt{\frac{X+Y}{\varepsilon_{k+1} k^3}}$$

$$\lesssim \frac{\log \left( 1/(m_k^x m_k^y) \right)}{k} + \sqrt{\frac{X+Y}{\varepsilon_{k+1} k^3}} = \mathcal{O} \left( \frac{(X+Y)^{\frac{1}{2}} \log \left( 1/(m_k^x m_k^y) \right)}{k^{\min\{1,(3-\alpha_\varepsilon)/2\}}} \right),$$

where in the last inequality, we use the fact that

$$\frac{(\varepsilon_k - \varepsilon_{k+1})}{\varepsilon_{k+1}} = \frac{k^{-\alpha_\varepsilon}}{(k+1)^{-\alpha_\varepsilon}} - 1 = (1 + 1/k)^{\alpha_\varepsilon} - 1 = \mathcal{O} \left( \frac{\alpha_\varepsilon}{k} \right),$$

by Taylor expansion. $\qquad\square$

# E. Last-iterate Convergence Rate in Expectation

**Theorem E.1.** *With the same condition as in Theorem 5.1, Algorithm 1 guarantees that*

$$\mathbb{E} \left[ \text{NEGap}(\mu^k, \nu^k) \right] = \widetilde{\mathcal{O}} \left( \left( (X+Y)^{\frac{1}{4}} H + \sqrt{(X^2 A + Y^2 B)} \right) k^{-\frac{1}{6}} \right).$$

*Proof.* With the same arguments as in the proof of Theorem 5.1, we have

$$D_\psi \left( \xi^{k+1,x}, \xi^{k+1} \right)$$
$$\leq (1 - \eta_k \varepsilon_k) D_\psi \left( \xi^{k,\star}, \xi^k \right) + \eta_k^2 \left( X \underline{\tau}_k + Y \bar{\tau}_k \right) + \eta_k^2 \left( X^2 A + Y^2 B \right) + \eta_k \rho_k + \eta_k \sigma_k + \omega_k$$
$$+ \eta_k^2 X A \varepsilon_k^2 \left( \log X + H \log (Ak) \right)^2 + \eta_k^2 Y B \varepsilon_k^2 \left( \log Y + H \log (Bk) \right)^2.$$

Taking conditional expectation $\mathbb{E}_{k-1}[\cdot]$ on both sides and by noticing the fact that $\mathbb{E}_{k-1}[\tau_k] < 0$, $\mathbb{E}_{k-1}[\rho_k] = 0$, and $\mathbb{E}_{k-1}[\sigma_k] = 0$, we have

$$\mathbb{E}_{k-1} \left[ D_\psi \left( \xi^{k+1,x}, \xi^{k+1} \right) \right]$$

$$\leq (1 - \eta_k \varepsilon_k) D_\psi \left( \xi^{k,\star}, \xi^k \right) + \eta_k^2 \left( X^2 A + Y^2 B \right) + \mathbb{E}_{k-1} \left[ \omega_k \right]$$
$$+ \eta_k^2 X A \varepsilon_k^2 \left( \log X + H \log \left( Ak \right) \right)^2 + \eta_k^2 Y B \varepsilon_k^2 \left( \log Y + H \log \left( Bk \right) \right)^2 .$$

Expanding the recursion in the above display leads to

$$\mathbb{E} \left[ D_\psi \left( \xi^{k+1,\star}, \xi^{k+1} \right) \right]$$
$$\leq \mathbb{E} \left[ \sum_{i=1}^k w_k^i \omega_i \right] + X A \left( \log X + H \log \left( Ak \right) \right)^2 \sum_{i=1}^k w_k^i \left( \eta_i \varepsilon_i \right)^2 + Y B \left( \log Y + H \log \left( Bk \right) \right)^2 \sum_{i=1}^k w_k^i \left( \eta_i \varepsilon_i \right)^2$$
$$+ \sum_{i=1}^k w_k^i \eta_i^2 \left( X^2 A + Y^2 B \right)$$
$$\leq (X + Y)^{\frac{1}{2}} \left( H \log(Ak) + H \log(Bk) \right) \left( \log X + H \log(Ak) + \log Y + H \log(Bk) \right)$$
$$\cdot \log(k) k^{- \min\{1, \frac{3}{2} - \frac{\alpha_\varepsilon}{2}\} - \alpha_\eta + \alpha_\varepsilon}$$
$$+ \left( X A \left( \log X + H \log \left( Ak \right) \right)^2 + Y B \left( \log Y + H \log \left( Bk \right) \right)^2 \right) k^{-\alpha_\eta - \alpha_\varepsilon} + \left( X^2 A + Y^2 B \right) k^{-\alpha_\eta + \alpha_\varepsilon}$$
$$= \widetilde{\mathcal{O}} \left( (X + Y)^{\frac{1}{2}} H^2 k^{- \min\{1, \frac{3}{2} - \frac{\alpha_\varepsilon}{2}\} + \alpha_\eta + \alpha_\varepsilon} + (X A + Y B) H^2 k^{-\alpha_\eta - \alpha_\varepsilon} + \left( X^2 A + Y^2 B \right) k^{-\alpha_\eta + \alpha_\varepsilon} \right) .$$

Hence,

$$\mathrm{NEGap}(\mu^k, \nu^k)$$
$$= \widetilde{\mathcal{O}} \left( \varepsilon_k H + \frac{XAH}{k} + \frac{YBH}{k} \right.$$
$$+ (X + Y) \left[ (X + Y)^{\frac{1}{4}} H k^{\left( - \min\{1, \frac{3}{2} - \frac{\alpha_\varepsilon}{2}\} + \alpha_\eta + \alpha_\varepsilon \right)/2} + \sqrt{(XA + YB)} + H k^{\frac{-\alpha_\eta - \alpha_\varepsilon}{2}} \right.$$
$$+ \left. \left. \sqrt{(X^2 A + Y^2 B)} k^{\frac{-\alpha_\eta + \alpha_\varepsilon}{2}} \right] \right)$$
$$= \widetilde{\mathcal{O}} \left( k^{-\frac{1}{6}} H + \frac{XAH}{k} + \frac{YBH}{k} + (X + Y) \left[ (X + Y)^{\frac{1}{4}} H k^{-\frac{1}{6}} + \sqrt{(XA + YB)} H k^{-\frac{1}{3}} \right. \right.$$
$$+ \left. \left. \sqrt{X^2 A + Y^2 B} k^{-\frac{1}{6}} \right] \right)$$
$$= \widetilde{\mathcal{O}} \left( (X + Y) \left[ (X + Y)^{\frac{1}{4}} H + \sqrt{(X^2 A + Y^2 B)} \right] k^{-\frac{1}{6}} \right) .$$

$\square$

# F. Last-iterate Convergence of OMD with Vanilla Negentropy

**Theorem F.1.** *If Algorithm 1 is adopted by both players and the vanilla negentropy $\psi(\mu) = \sum_{h, x_h, a_h} \mu_{1:h}(x_h, a_h) \log \mu_{1:h}(x_h, a_h)$ is used, by setting $\alpha_\eta = 5/8$, $\alpha_\gamma = 3/8$ and $\alpha_\varepsilon = 1/8$, for any $k \geqslant 1$, with probability at least $1 - \widetilde{\mathcal{O}}(\delta)$, it holds that*

$$\mathrm{NEGap}(\mu^k, \nu^k) = \widetilde{\mathcal{O}} \left( (XA + YB) H k^{-\frac{1}{8}} \right) .$$

*Proof.* The proof is mostly similar to that of Theorem 5.1 and hence we only present the key steps that differ from those in the proof of Theorem 5.1.

To start with, since vanilla negentropy is 1-strongly-convex with respective to the $\ell_2$-norm (see, *e.g.*, Lemma 11 of Lee et al. (2021)), following a similar analysis in the proof of Theorem 5.1, we have

$$\mathrm{NEGap} \left( \xi^k \right) \leq \mathrm{NEGap} \left( \xi^{k,\star} \right) + \sqrt{XA} \left\| \mu^k - \mu^{k,\star} \right\|_2 + \sqrt{YB} \left\| \nu^k - \nu^{k,\star} \right\|_2$$
$$\lesssim \mathrm{NEGap} \left( \xi^{k,\star} \right) + \left( \sqrt{XA} + \sqrt{YB} \right) \sqrt{D_\psi \left( \xi^{k,\star}, \xi^k \right)} .$$

When the vanilla negentropy is used, building on a similar analysis as in the proof of Lemma C.1, it holds that

$$\text{NEGap}(\xi^{k,\star}) = \widetilde{\mathcal{O}}\left(\varepsilon_k(X+Y) + \frac{XAH}{k} + \frac{YBH}{k}\right) . \tag{42}$$

To bound $D_\psi\left(\xi^{k,\star}, \xi^k\right)$, by similar analysis of Lemma C.2, one can see that

$$
\begin{aligned}
&D_\psi\left(\xi^{k+1,\star}, \xi^{k+1}\right) \\
&\leq \underbrace{\sum_{i=1}^k w_k^i \eta_i \rho_i}_{\textbf{Term 1}} + \underbrace{\sum_{i=1}^k w_k^i \eta_i \sigma_i}_{\textbf{Term 2}} + \underbrace{XA(\log^2 m_k^x)\sum_{i=1}^k w_k^i \left(\eta_i \varepsilon_i\right)^2 + YB(\log^2 m_k^y)\sum_{i=1}^k w_k^i \left(\eta_i \varepsilon_i\right)^2}_{\textbf{Term 3}} \\
&+ \underbrace{\sum_{i=1}^k w_k^i \eta_i^2 \left(\underline{\tau}_i + \bar{\tau}_i\right)}_{\textbf{Term 4}} + \underbrace{\sum_{i=1}^k w_k^i \eta_i^2 \left(XA + YB\right)}_{\textbf{Term 5}} + \underbrace{\sum_{i=1}^k w_k^i \omega_i}_{\textbf{Term 6}} .
\end{aligned}
\tag{43}
$$

Note that **Term 1** through **Term 3** are the same as in the case of using virtual transition-weighted negentropy and can be bounded by Lemma C.3 - Lemma C.5, respectively. For **Term 4** and **Term 5**, by similar analysis of Lemma C.6 and Lemma C.7, we have

$$\textbf{Term 4} \leq k^{\alpha_\gamma - 2\alpha_\eta} \log\left(1/\delta\right) ,$$

and

$$\textbf{Term 5} \lesssim (XA + YB)\, k^{-\alpha_\eta + \alpha_\varepsilon} .$$

On the other hand, using again the fact that vanilla negentropy is 1-strongly-convex with respective to the $\ell_2$-norm and the analysis of Lemma C.8, we can show that

$$\textbf{Term 6} \lesssim (XA + YB)\log^2\left(1/\left(m_k^x m_k^y\right)\right)\log(k) k^{-\min\{1,(3-\alpha_\varepsilon)/2\}+\alpha_\eta+\alpha_\epsilon} .$$

The proof is concluded by putting all the above together. $\qquad\square$

# G. Auxiliary Lemmas

**Lemma G.1.** *Let $\{c_i\}_{i=1}^k$ be fixed positive numbers. Then with probability at least $1 - \delta$, it holds that*

$$\sum_{i=1}^k c_i \left\langle \mu^i, \ell^{i,x} - \widehat{\ell}^{i,x} \right\rangle \leq XA\sum_{i=1}^k c_i \gamma_i + H\sqrt{2\sum_{i=1}^k c_i^2 \log\frac{1}{\delta}} .$$

*Proof.* To begin with, notice that

$$\sum_{i=1}^k c_i \left\langle \mu^i, \ell^{i,x} - \widehat{\ell}^{i,x} \right\rangle = \sum_{i=1}^k c_i \left\langle \mu^i, \ell^{i,x} - \mathbb{E}_{i-1}\left[\widehat{\ell}^{i,x}\right] \right\rangle + \sum_{i=1}^k c_i \left\langle \mu^i, \mathbb{E}_{i-1}\left[\widehat{\ell}^{i,x}\right] - \widehat{\ell}^{i,x} \right\rangle .$$

For the first part, we have

$$
\begin{aligned}
&\sum_{i=1}^k c_i \left\langle \mu^i, \ell^{i,x} - \mathbb{E}_{i-1}\left[\widehat{\ell}^{i,x}\right] \right\rangle \\
&= \sum_{i=1}^k c_i \sum_{h,x_h,a_h} \mu_{1:h}^i\left(x_h, a_h\right) \ell^{i,x}\left(x_h, a_h\right) \left(1 - \frac{\mu_{1:h}^i\left(x_h, a_h\right)}{\mu_{1:h}^i\left(x_h, a_h\right) + \gamma_i}\right)
\end{aligned}
$$

$$\leq \sum_{i=1}^{k} c_i \gamma_i \sum_{h,x_h,a_h} \ell^{i,x}(x_h, a_h)$$

$$\leq \sum_{i=1}^{k} c_i \gamma_i X A,$$

where the last inequality comes from $\ell^{i,x}(x_h, a_h) \leq 1$ for all $(x_h, a_h) \in \mathcal{X} \times \mathcal{A}$.

For the second part, taking $\delta = \exp\left(-\varepsilon^2 / \left(2 \sum_{i=1}^{k} c_i^2 H^2\right)\right)$, $\varepsilon = \sqrt{2 \sum_{i=1}^{k} c_i^2 H^2 \log(1/\delta)}$ and using Azuma-Hoeffding inequality, it holds with probability at least $1 - \delta$ that

$$\sum_{i=1}^{k} c_i \left\langle \mu^i, \mathbb{E}_{i-1}\left[\widehat{\ell}^{i,x}\right] - \widehat{\ell}^{i,x} \right\rangle \leq \sqrt{2 \sum_{i=1}^{k} c_i^2 H^2 \log\left(\frac{1}{\delta}\right)}.$$

The proof is concluded by combining the upper bounds of the two parts above. $\qquad\square$

**Lemma G.2.** *Let* $\delta \in (0,1)$ *and* $\{\gamma_i\}_{i=1}^{k} \in (0, +\infty)^k$. *Fix* $h \in [H]$. *For any coefficient sequence* $\{c_i\}_{i=1}^{k}$ *s.t.* $c_i \in [0, 2\gamma_i]^{XA}$ *is* $\mathcal{F}_{i-1}$ *- measurable, with probability* $1 - \delta$, *we have*

$$\sum_{i=1}^{k} w_i \left\langle c_i, \widehat{\ell}_i - \ell_i \right\rangle \leq \max_{1 \leq i \leq k} w_i \log \frac{1}{\delta}.$$

*Proof.* Define $w = \max_{1 \leq i \leq k} w_i$. Hence

$$
\begin{aligned}
w^i \widehat{\ell}^i(x_h, a_h) &= \frac{w_i \mathbb{I}_{i,h}\{x_h, a_h\}(1 - r_h^i)}{\mu_{1:h}^i(x_h, a_h) + r_i} \\
&\leq \frac{w_i \mathbb{I}_{i,h}\{x_h, a_h\}(1 - r_h^i)}{\mu_{1,h}^i(x_h, a_h) + r_i w_i (1 - r_h^i)\mathbb{I}_{i,h}\{x_h, a_h\}/w} \\
&= \frac{w}{2\gamma_i} \cdot \frac{2\gamma_i w_i(1 - r_h^i)\mathbb{I}_{i,h}\{x_h, a_h\}/(w\mu_{1:h}^i(x_h, a_h))}{1 + \gamma_i w_i (1 - r_h^i)\mathbb{I}_{i,h}\{x_h, a_h\}/(w\mu_{1:h}^i(x_h, a_h))} \\
&\leq \frac{w}{2\gamma_i} \log\left(1 + \frac{2\gamma_i w_i(1 - r_h^i)\mathbb{I}_{i,h}\{x_h, a_h\}}{w\mu_{1:h}^i(x_h, a_h)}\right).
\end{aligned}
$$

Denote by $\widehat{S}_h^i = \frac{w_i}{w}\left\langle c_i, \widehat{\ell}_h^i\right\rangle$, $S_h^i = \frac{w_i}{w}\left\langle c_i, \ell_h^i\right\rangle$. Then

$$
\begin{aligned}
\mathbb{E}_{i-1}\left[\exp\left(\widehat{S}^i\right)\right] &\leq \mathbb{E}_{i-1}\left[\exp\left(\sum_{(x_h,a_h)\in\mathcal{X}\times\mathcal{A}} \frac{c_i(x_h, a_h)}{2\gamma_i}\log\left(1 + \frac{2\gamma_i w_i(1 - r_h^i)\mathbb{I}_{i,h}\{x_h, a_h\}}{w\mu_{1:h}^i(x_h, a_h)}\right)\right)\right] \\
&\leq \mathbb{E}_{i-1}\left[\prod_{(x_h,a_h)\in\mathcal{X}\times\mathcal{A}}\left(1 + \frac{c_i(x_h, a_h)w_i(1 - r_h^i)\mathbb{I}_{i,h}\{x_h, a_h\}}{w\mu_{1:h}^i(x_h, a_h)}\right)\right] \\
&= \mathbb{E}_{i-1}\left[1 + \sum_{(x_h,a_h)\in\mathcal{X}\times\mathcal{A}} \frac{c_i(x_h, a_h)w_i(1 - r_h^i)\mathbb{I}_{i,h}\{x_h, a_h\}}{w\mu_{1:h}^i(x_h, a_h)}\right] \\
&= 1 + S_h^i \leq \exp\left(S_h^i\right).
\end{aligned}
$$

Finally, one can see that

$$\mathbb{E}\left[\sum_{i=1}^{k}\left(\widehat{S}_h^i - S_h^i\right) \geq \log\frac{1}{\delta}\right] = \mathbb{E}\left[\exp\left(\sum_{i=1}^{k}\left(\widehat{S}_h^i - S_h^i\right)\right) \geq \frac{1}{\delta}\right]$$

$$\leq \delta \mathbb{E}\left[\exp\left(\sum_{i=1}^{k}\left(\widehat{S}_h^i - S_h^i\right)\right)\right]$$

$$= \delta \mathbb{E}\left[\left[\mathbb{E}_{k-1}\left[\exp\left(\sum_{i=1}^{k}\left(\widehat{S}_h^i - S_h^i\right)\right)\right]\right]\right]$$

$$= \delta \mathbb{E}\left[\exp\left(\sum_{i=1}^{k-1}\left(\widehat{S}_h^i - S_h^i\right)\right)\left[\mathbb{E}_{k-1}\left[\exp\left(\widehat{S}_h^k - S_h^k\right)\right]\right]\right] \leq \ldots \leq \delta .$$

$\square$

**Lemma G.3.** *Let $\{c_i\}_{i=1}^{k}$ be fixed positive numbers. Fix $h \in [H]$. Then $\forall$ sequence $\{q_i\}_{i=1}^{k} \in [0,1]^{XA}$ s.t. $q^i$ is $\mathcal{F}_{i-1}$-measurable, with probability at least $1 - \delta$,*

$$\sum_{i=1}^{k} c_i \left\langle q_i, \widehat{\ell}_h^i - \ell_h^i \right\rangle \leq \max_{1 \leq i \leq k} \frac{c_i}{\gamma_k} \log\left(\frac{1}{\delta}\right) .$$

*Proof.* Noticing that $\{\gamma_i\}_{i=1}^{k}$ is decreasing and $\|q^i\|_\infty \leq 1$, applying Lemma G.2, we arrive at

$$\sum_{i=1}^{k} c_i \left\langle q^i, \widehat{\ell}_h^i - \ell_h^i \right\rangle = \sum_{i=1}^{k} \frac{c_i}{2\gamma_i} \left\langle 2\gamma_i q^i, \widehat{\ell}_h^i - \ell_h^i \right\rangle \leq \max_{1 \leq i \leq k} \frac{c_i}{\gamma_k} \log\left(\frac{1}{\delta}\right) .$$

$\square$

**Lemma G.4** (Lemma 1 of Cai et al. (2023)). *Let $0 < h < 1, 0 \leq k \leq 2$, and let $t \geqslant \left(\frac{24}{1-h} \ln \frac{12}{1-h}\right)^{\frac{1}{1-h}}$. Then*

$$\sum_{i=1}^{k}\left(i^{-k} \prod_{j=i+1}^{k}\left(1 - j^{-h}\right)\right) \leq 9 \ln(t) t^{-k+h} .$$

**Lemma G.5** (Lemma 2 of Cai et al. (2023)). *Let $0 < h < 1, 0 \leq k \leq 2$, and let $t \geqslant \left(\frac{24}{1-h} \ln \frac{12}{1-h}\right)^{\frac{1}{1-h}}$. Then*

$$\max_{1 \leq i \leq t}\left(i^{-k} \prod_{j=i+1}^{k}\left(1 - j^{-h}\right)\right) \leq 4 t^{-k} .$$

**Lemma G.6** (Lemma 12 of Cai et al. (2023)). *For all $x \in (0,1)$ and $y > 0$, it holds that $x^{1-y} - x \leq -y x^{1-y} \ln x$.*

**Lemma G.7** (Lemma 20 of Bai et al. (2020)). *Let $c_1, c_2, \ldots, c_t$ be fixed positive numbers. Then with probability at least $1 - \delta$,*

$$\sum_{i=1}^{k} c_i \left\langle x_i, \ell_i - \widehat{\ell}_i \right\rangle = \mathcal{O}\left(A \sum_{i=1}^{k} \beta_i c_i + \sqrt{\ln(A/\delta) \sum_{i=1}^{k} c_i^2}\right) .$$

## H. Proof of Lower Bound of Last-iterate Convergence

*Proof of Theorem 5.6.* Let $\text{NEGap}_k := \text{NEGap}\left(\mu^k, \nu^k\right)$ with $\left(\mu^k, \nu^k\right)$ as the policy profile generated by some algorithm Alg. Suppose that Alg leans the IIEFG with the last-iterate convergence rate of $\text{NEGap}_k = \Theta\left(f(X, A) k^{-\alpha}\right)$ for some $\alpha \in (0, 1)$, where $f^{\text{Alg}}(X, A)$ denotes the polynomial dependence on $X$ and $A$ of $\text{NEGap}_k$.

Fix some $K \geqslant \max(XA, YB)$. Consider the regret defined as follows (Kozuno et al., 2021; Bai et al., 2022; Fiegel et al., 2023):

$$\text{Reg}_K(\text{Alg}) = \sup_{\mu \in \Pi_{\max}} \sum_{k=1}^{K} \left\langle \mu^k - \mu, \boldsymbol{G} \nu^k \right\rangle ,$$

where $\{\nu^k\}_{k\in[K]}$ is potentially generated by an adversary. Then, one can deduce that

$$\text{Reg}_K(\text{Alg}) = \sup_{\mu\in\Pi_{\max}} \sum_{k=1}^{K} \langle \mu_k - \mu, \boldsymbol{G}\nu_k \rangle \tag{44}$$

$$\leq \sum_{k=1}^{K} \sup_{\mu\in\Pi_{\max}} \langle \mu_k - \mu, \boldsymbol{G}\nu_k \rangle$$

$$= \sum_{k=1}^{K} \sup_{\mu\in\Pi_{\max}} \mu_k^\top \boldsymbol{G}\nu_k - \mu^\top \boldsymbol{G}\nu_k$$

$$\leq \sum_{k=1}^{K} \sup_{\mu\in\Pi_{\max}, \nu\in\Pi_{\min}} \mu_k^\top \boldsymbol{G}\nu - \mu^\top \boldsymbol{G}\nu_k$$

$$= \sum_{k=1}^{K} \text{NEGap}_k$$

$$= \Theta\left( f(X,A) \sum_{k=1}^{K} k^{-\alpha} \right)$$

$$= \Theta\left( f(X,A)K^{1-\alpha} \right). \tag{45}$$

On the other hand, by Theorem 6 of Bai et al. (2022) (see also Theorem 3.1 of Fiegel et al. (2023)), we have

$$\text{Reg}_K(\text{Alg}) \geqslant \Omega(\sqrt{AXK}). \tag{46}$$

Combining Eq. (44) and Eq. (46), we have

$$\Omega(\sqrt{AXK}) \leq \Theta\left( f(X,A)K^{1-\alpha} \right).$$

We now further consider the following three cases:

- If $\alpha > \frac{1}{2}$, then $\sqrt{AX} \leq f(X,A)K^{\frac{1}{2}-\alpha}$. However, this does not hold for any $f$, when $K$ is large enough;

- If $\alpha = \frac{1}{2}$, it must hold that $\sqrt{AX} \leq f(X,A)$;

- If $\alpha < \frac{1}{2}$, then $\sqrt{AX} \leq f(X,A)K^{\frac{1}{2}-\alpha}$. This holds for all $f$, including $f(X,A) = 1$ when $K$ is large enough. In this case, the "minimal" $f$ is $f(X,A) = 1$, implying that the minimal possible convergence rate of $\text{NEGap}_k$ in this case is $\text{NEGap}_k = \Theta\left( k^{-\alpha} \right)$.

Taking the above three cases into account, the minimal possible convergence rate is

$$\min\left\{ \Theta\left( \sqrt{XA}k^{-\frac{1}{2}} \right), \Theta\left( k^{-\alpha} \right) \right\} \quad (\alpha > \frac{1}{2})$$

$$= \Theta\left( \sqrt{XA}k^{-\frac{1}{2}} \right).$$

Analogously, we can prove that $\text{NEGap}_k \geq \Theta(\sqrt{YB}k^{-\frac{1}{2}})$. Therefore, we have

$$\text{NEGap}_k \geq \Theta\left( \left( \sqrt{XA} + \sqrt{YB} \right) k^{-\frac{1}{2}} \right).$$

The proof is concluded by noticing that the above holds for all algorithms. $\qquad\square$

# I. Experiments

In this section, we present the empirical evaluations of our Algorithm 1. [3] Since we are not aware of any other algorithm that can also learn the (approximate) NE policy profile in IIEFGs with provable *last-iterate convergence* guarantees under bandit feedback, we compare our algorithm against previous algorithms that converge to the (approximate) NE policy profile in IIEFGs with only *average-iterate convergence* guarantees including IXOMD (Kozuno et al., 2021), BalancedOMD (Bai et al., 2022) and BalancedFTRL (Fiegel et al., 2023). Since these algorithms are only devised to obtain the average-iterate convergence for learning IIEFGs, the last-iterate convergence of these algorithms for learning IIEFGs is not theoretically guaranteed.

**Environments**   We consider four standard IIEFG instances including Lewis Signaling, Kuhn Poker (Kuhn, 1950), Leduc Poker (Southey et al., 2012) and Liars Dice. All the implementations of these games are from the OpenSpiel library (Lanctot et al., 2019).

**Implementation Details**   As mentioned in Section 4.1, for our Algorithm 1, instead of operating the update OMD in the constrained set $\Pi_{\max}^k$, we clip the loss estimator and then perform the update of OMD in the whole $\Pi_{\max}$ to compute an approximate update using the closed-form solution in Algorithm 3. The clipping estimator $\zeta^{k+1}$ in Eq. (13) is set as $\zeta^{k+1} = 1 \times 10^{-10}$. We adopt the implementation of all the baselines by Fiegel et al. (2023). Besides, we consider a (logarithmic) grid search on the learning rates for all the algorithms, following Fiegel et al. (2023). All the experiments are conducted on a server with an Intel Xeon Gold CPU and 251GiB system memory. The running of all the algorithms including our algorithm costs approximately 10 hours, 12 hours, 13 hours, and 16 hours on Lewis Signaling, Kuhn Poker, Leduc Poker, and Liars Dice, respectively.

**Results**   The experimental results are shown in Figure 1. Our algorithm obtains the best or the competitive performance across all four IIEFG instances. In particular, our algorithm converges faster than all the baseline algorithms on Kuhn Poker and Liars Dice and also converges as fast as the empirically best baseline algorithm on Lewis Signaling and Leduc Poker. Though some baseline algorithms work relatively well on some game instances, we would like to note again that these algorithms are not theoretically guaranteed to converge to the NE policy profile with the last-iterate convergence. We speculate that this might also be the reason why some baseline algorithms perform relatively well in some instances but poorly in the remaining ones. For instance, the BalancedFTRL algorithm performs well on Leduc Poker while converging very slowly on Kuhn Poker. Analogously, BalancedOMD converges relatively well on Kuhn Poker and Leduc Poker but converges the most slowly on Liars Dice.

Moreover, in general, it appears that the advantage of our algorithm becomes more pronounced in IIEFG instances with larger infoset spaces $\mathcal{X}$ (and action spaces $\mathcal{A}$) over previous algorithms. This observation aligns with the intuition that in such instances, the baseline algorithms, which solely have average-iterate convergence theoretical guarantees, face greater difficulty in achieving last-iterate convergence to the NE. This challenge may arise because these algorithms are more susceptible to getting stuck in suboptimal policy profiles, due to lack of the last-iterate convergence theoretical guarantees.

---

[3]Codes of the experiments are available at https://github.com/ColoeredGalaxy/Last_ite_Convergence_in_EFGs.

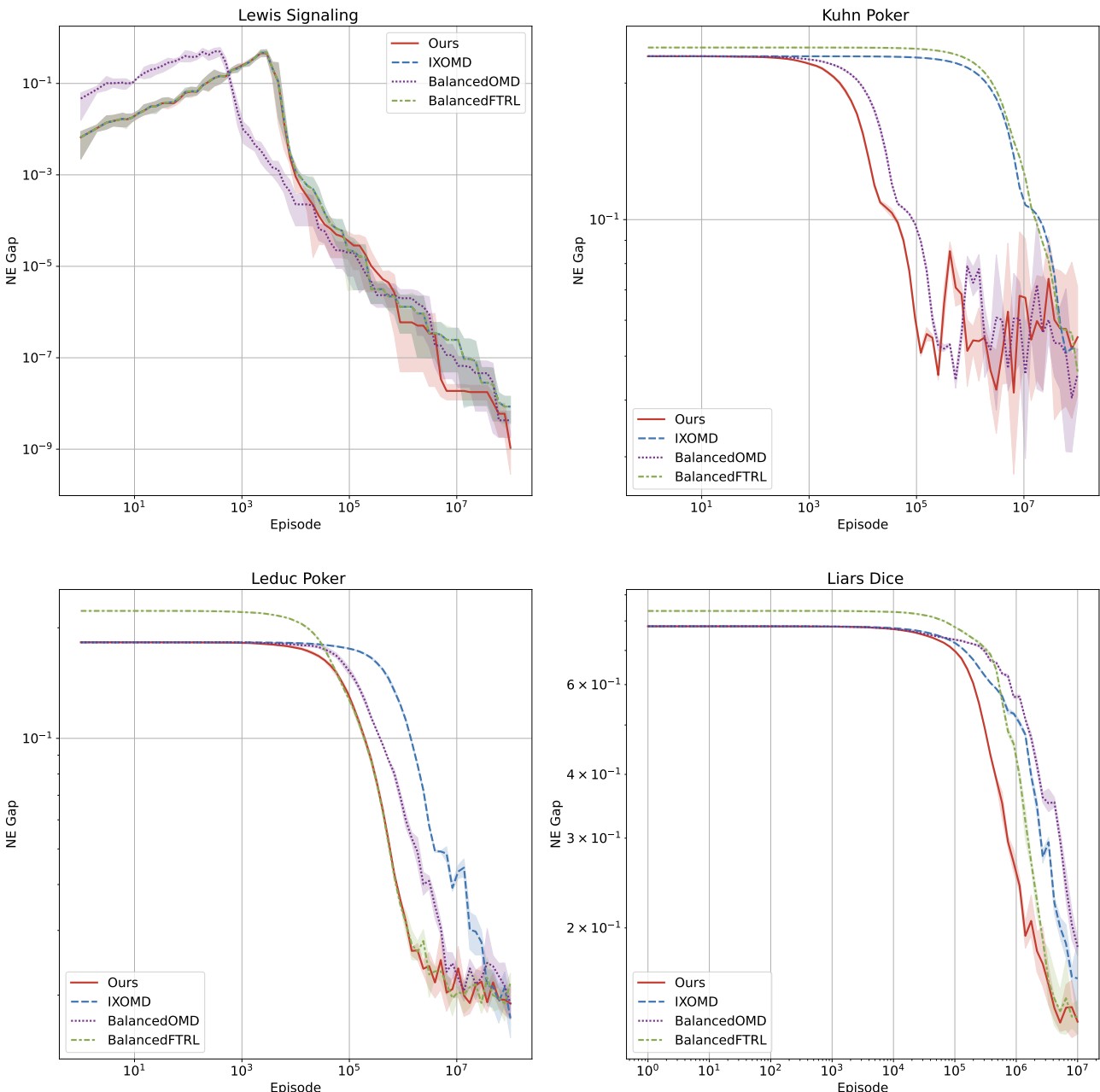

Figure 1. We present experimental results of our Algorithm 1 in comparison with IXOMD (Kozuno et al., 2021), BalancedOMD (Bai et al., 2022), and BalancedFTRL (Fiegel et al., 2023). The curves depict the last-iterate convergence of the NE gap, as defined in Eq. (1), versus the number of episodes.

