# OpenReview forum: "Learning Imperfect Information Extensive-form Games with Last-iterate Convergence under Bandit Feedback"
_ICML.cc/2025/Conference — ICML 2025 poster_

### Official Review · Reviewer_wkY8 · 2025-02-14

**Overall Recommendation:** 2

**Summary:**

The authors propose an efficient algorithm (namely, with closed form solution update) which attains a last-iterate convergence rate of order $K^{-1/8}$ when run in self-play and when only bandit feedback is available. They also provide a lower-bound on the convergence which does not match the rate attained by the algorithm.

**Claims And Evidence:**

The result seems convincing and reasonable.

**Essential References Not Discussed:**

The related references are cited in the work.

**Experimental Designs Or Analyses:**

I checked the experimental results in the Appendix.

**Methods And Evaluation Criteria:**

The proposed method and evaluation criteria are standard.

**Other Comments Or Suggestions:**

No additional comments.

**Other Strengths And Weaknesses:**

The main strength of the is work is that the authors show the first result on last-iterate convergence in EFGs with bandit feedback and uncoupled dynamic. Nonetheless, I have concerns regards the technical novelty of this work. Specifically, the techniques seem to me mainly adapted from (Cai et al. 2023), which established almost identical result for matrix games with bandit feedback. Indeed, the are some challenges in dealing with EFGs; nevertheless, the techniques employed to generalise (Cai et al. 2023) to EFGs are mainly adapted from (Kozuno et al. 2021) (e.g., notice that, closed form solution for OMD is not surprising in EFGs, when proper distance generating function are employed).

Overall, I believe that the contribution of this paper is not enough to meet the acceptance bar of a conference such as ICML.

**Questions For Authors:**

Which are the fundamental differences of this work with respect to (Cai et al. 2023)? I am willing to increase my score if the answers are pretty positive.

**Relation To Broader Scientific Literature:**

The only contribution of this works is to extend previous results in matrix games to extensive form games.

**Theoretical Claims:**

I did not check the correctness of the proof, in details.

---

> ### Author Rebuttal · Authors · 2025-03-30
>
> We thank the reviewer for the valuable comments. Our response to each question is provided below.
>
> **Q1. Specifically, the techniques seem to me ... when proper distance generating function are employed).**
>
> Thank you for this comment. Indeed, our analysis scheme is inspired by [1], as we have mentioned in Remark 5.5 of our paper. However, we would also like to remark that a key distinction between our work and [1] is that [1] uses the *vanilla negentropy regularizer* while we use a *virtual transition-weighted negentropy regularizer*. Simply using the vanilla negentropy regularizer in our problem can still guarantee a finite-time last-iterate convergence result as mentioned in Remark 5.4 of our paper, but doing so will not be able to obtain computationally efficient approximate policy updates. In contrast, with the leverage of the virtual transition-weighted negentropy regularizer, our algorithm simultaneously permits efficient approximate policy updates (please see the end of Section 4.1 and Appendix A in our paper for details) and guarantees a meaningful last-iterate convergence.
>
> Besides, we fully agree that closed-form updates for OMD are common in IIEFGs. And it is precisely for this reason that we would like to make our algorithm computationally efficient as well. Nevertheless, we would like to note that, though dilated negentropy is common in existing literature studying IIEFGs (say, [2,3,4]), we did not use the dilated negentropy as the regularizer in our algorithm. Though it is well known that dilated negentropy admits efficient closed-form policy updates, using dilated negentropy in our problem can only lead to a vacuous convergence rate that is exponentially large in $X$, as illustrated in Section 4.1 of our paper. In contrast, as aforementioned, we leverage the negentropy weighted by the virtual transition over the infoset-action spaces. Importantly, this virtual transition is specifically designed to maximize the minimum sequence-form "visitation probability" of all the infoset-action pairs, so as to ensure a last-iterate convergence with meaningful dependence on $X$. Our virtual transition is directly computed by our Algorithm 2, and the construction of such a virtual transition is never exploited in existing literature for studying IIEFGs, let alone in [1]. Therefore, we believe our algorithmic design as well as the result is meaningful and valuable to the literature.
>
> We hope the above explanations would be helpful to address the concerns of the reviewer about our technical contributions, and we will explicitly incorporate the above discussions into the revision of our paper for better clarity.
>
> **Q2. Which are the fundamental differences of this work with respect to (Cai et al. 2023)?**
>
> Please see our response to the question above.
>
>
>
> [1] Cai et al. Uncoupled and convergent learning in two-player zero-sum markov games with bandit feedback. NeurIPS, 2023.
>
> [2] Hoda et al. Smoothing techniques for computing nash equilibria of sequential games. MOR, 2010.
>
> [3] Kroer et al. Faster First-Order Methods for Extensive-Form Game Solving. EC, 2015.
>
> [4] Kozuno et al. Learning in two-player zero-sum partially observable markov games with perfect recall. NeurIPS, 2021

---

> > ### Comment · Reviewer_wkY8 · 2025-04-01
> >
> > I would like to thank the Authors for the detailed response. Nonetheless, after reading the rebuttal and the answer to Question 2 (Q2) of Reviewer T2uG, I am still not convinced that the paper has sufficient technical contribution. Thus, for now, I will keep my current score.

---

> > > ### Author Response · Authors · 2025-04-03
> > >
> > > We thank the reviewer for the prompt response.

---

### Official Review · Reviewer_T2uG · 2025-03-12

**Overall Recommendation:** 4

**Summary:**

The paper proposes an algorithm for two-player zero-sum extensive-form games (2p0g) with bandit feedback. Under the self-play setting, the last-iterate of a profile computed by the proposed algorithm converges to the Nash equilibrium in a rate of $k^{-1/8}$ (or $k^{-1/6}$ in expectation). The main innovation is the use of a negentropy regularization instead of dilated entropy regularization frequently used in other algorithms for 2p0g, combined with a novel technique called virtual transition.

## update after rebuttal

I thank the authors for the rebuttal responses. I am convinced by the authors' explanation and raised my score to "Accept".

**Claims And Evidence:**

The following is a list of claims.
1. The first algorithm for 2p0g under the bandit feedback setting with finite-time last-iterate convergence guarantee.
2. Their algorithm still admits a closed form solution for policy updates over the full policy space.
3. Their algorithm does not require any communication or coordination between the two players and is model-free
4. A sample complexity lower bound for the last-iterate convergence rate.

Regarding 1, I would like to mention that there exists an algorithm for 2p0g under the __noisy-feedback setting__ with finite-time last-iterate convergence guarantee. Indeed, [Abe et al. (2024)](https://proceedings.mlr.press/v235/abe24a.html) provides one. I recommend the authors to discuss it, especially why deriving a sample complexity bound like the one in the paper is difficult to derive.

Regarding 2 and 3, they are supported by Appendix A and Algorithm 1, respectively.

Regarding 4, the stated lower bound is correct and coincides with one obtained previously in Fiegel et al. (2023).

**Essential References Not Discussed:**

Maybe [Abe et al. (2024)](https://proceedings.mlr.press/v235/abe24a.html).

**Experimental Designs Or Analyses:**

Yes.

**Methods And Evaluation Criteria:**

Experiments are conducted in standard games, as shown in Figure 1, which compares NE gap of different algorithms after different episodes.

**Other Comments Or Suggestions:**

As noted above, Abe et al. (2024) provides a similar result, and it is nice to cite and discuss it. In addition, it would be nice to have some illuminating example and discussion on why virtual transition is necessary.

**Other Strengths And Weaknesses:**

As I noted above, a similar result is obtained in [Abe et al. (2024)](https://proceedings.mlr.press/v235/abe24a.html), that is, an algorithm with finite-time last-iterate convergence guarantee under the noisy feedback setting. However, they do not specifically consider 2p0g and issues related to loss estimates. This paper picks up 2p0g and handles bias and variance trade-off resulting in a result closer to a practical setting. Furthermore, the paper improves the existing convergence rate result of [Abe et al. (2024)](https://proceedings.mlr.press/v235/abe24a.html) from $\mathcal{O}(k^{-1/10})$ to $\mathcal{O}(k^{-1/8})$ with a completely different idea.

Regarding the lower bound, it seems it has been already shown in Fiegel et al. (2023).

The reason why I recommend weak acceptance is that the derived bound is not close to a lower bound, and it is not really clear why virtual transition is necessary.

**Questions For Authors:**

Q1. Would you explain how the lower bound in this paper differs from the one in Fiegel et al. (2023)? Since the latter one does not assume algorithm to be used, its lower bound applies to the setting considered in this paper too. Also, while their theorems states only sample complexity, they actually derive rate lower bounds in their appendix and use them to derive sample complexity lower bounds.

Q2. This paper states that "extracting (tree structure) from one traversal on the game tree" is easy, quoting Bai et al. (2022). However, I really did not understand what exactly this means (even when I read Bai et al. (2022) before). What does one traversal mean?

**Relation To Broader Scientific Literature:**

The paper contributes to scientific fields related to sequential decision making since the paper deepens the understanding of solving 2p0g with algorithms whose last iterate converges. It is currently an active research area.

**Theoretical Claims:**

No, but all theoretical results seem to be reasonable.

---

> ### Author Rebuttal · Authors · 2025-03-30
>
> We thank the reviewer for the valuable comments and suggestions. Our response to each question is provided below.
>
> **Q1. Additional References.**
>
> Thank you for referring to this! We compare our work with some notable works studying achieving last-iterate convergence in games with noisy feedback [1,2,3]. Specifically, [1,2,3] establish algorithms for solving two-player zero-sum matrix games or multi-player monotone games with noisy gradient feedback, where the noisy feedback for all the actions is observable. In contrast, we study learning IIEFGs with bandit feedback. That is, only the rewards of the experienced infoset-action pairs are revealed to the players. For the infoset-action pairs that are not experienced in one episode, no information regarding them is revealed, not even the noisy rewards. Hence, their algorithms are not directly applicable to our problem, and our results might not be directly comparable to theirs. Actually, in Sec. 8 of [2], the authors say that bandit feedback is a scheme with more limited feedback, and in Sec. 8 of [3], it is also mentioned that extending their results from the noisy feedback setting to the bandit feedback is a challenging future direction.
>
> We will explicitly incorporate the above discussions in the revision of our paper for completeness.
>
> **Q2. "it would be nice to ... why virtual transition is necessary."**
>
> Thanks for this comment! The primary effect of using a virtual transition-weighted negentropy regularizer is to ensure the algorithm can be approximately updated in a computationally efficient manner. Indeed, using a vanilla negentropy regularizer without the virtual transition can also lead to a finite-time last-iterate convergence for our problem, but in this way, the policy cannot be efficiently updated, as illustrated in Remark 5.4.
>
> Particularly, in matrix games, the policy space is simply the probability simplex over all the actions, and it is well known that using OMD/FTRL with vanilla negentropy regularizer admits a closed-form multiplicative update (see, e.g., Chap. 28 of [4]). However, in general IIEFGs, this is no longer the case since each sequence-form policy is not a probability measure over infoset-action space. Fortunately, the inner products between sequence-form virtual transitions and sequence-form policies are still probability measures over infoset-action space. Therefore, using a virtual transition-weighted negentropy regularizer permits efficient policy updates (please see Prop. A.1 in Appendix A). On the other hand, the downside of using a virtual transition-weighted negentropy regularizer is that it will enlarge the stability term (approximately) by a factor of $\frac{1}{p_{1: h}^x(x_h)}$ (please see Lem. D.1 in Appendix D). Therefore, it necessitates choosing a good virtual transition with $p_{1: h}^x(x_h)$ well lower-bounded, which is guaranteed by our Algorithm 2 (please see Lem. B.1 for details).
>
> We will incorporate the above explanations in our revised paper for clarity.
>
> **Q3. Lower Bound.**
>
> The main difference is that our lower bound only applies to the class of algorithms with last-iterate convergence. Technically, as shown in Appendix H, the proof idea of our lower bound is that any algorithm with $\Theta(k^{-\alpha})$ ($\alpha\in(0,1)$) last-iterate convergence can also guarantee a $\Theta(K^{1-\alpha})$ regret. Thus, we can prove the lower bound for the last-iterate convergence by contradiction using the existing regret minimization lower bound of $\Omega(\sqrt{XAK})$ in learning IIEFGs with bandit feedback (say, Thm. 6 in [5];  Thm. 3.1 in [6]). Our reduction is specifically designed for the class of algorithms with last-iterate convergence and does not directly work for the class of algorithms only with average-iterate convergence. On the other hand, we note that the hard instance used in our lower bound of last-iterate convergence is the same as that in [6], as the hard instances in the proofs of lower bounds of regret minimization and sample complexity are the same [5,6].
>
> We will further clarify this in the revision of our paper.
>
> **Q4. "What does one traversal mean?"**
>
> Due to the perfect recall condition, all the infoset-action pairs $\mathcal{X}\times\mathcal{A}$ across different steps $h$ form a tree. This tree can be constructed by performing the search (e.g., DFS, BFS) over all the infoset-action pairs one time.
>
> We will include the above explanations in the revision of our paper.
>
> [1] Abe et al. Last-iterate convergence with full and noisy feedback in two-player zero-sum games. AISTATS, 23.
>
> [2] Abe et al. Adaptively Perturbed Mirror Descent for Learning in Games. ICML, 24.
>
> [3] Abe et al. Boosting Perturbed Gradient Ascent for Last-Iterate Convergence in Games. ICLR, 25.
>
> [4] Lattimore et al. Bandit Algorithms. 20.
>
> [5] Bai et al. Near-optimal learning of extensive-form games with imperfect information. ICML, 22.
>
> [6] Fiegel et al. Adapting to game trees in zero-sum imperfect information game. ICML, 23.

---

> > ### Comment · Reviewer_T2uG · 2025-04-03
> >
> > First of all, I would like to thank the authors for the rebuttal comments.
> >
> > > The main difference is that our lower bound only applies to the class of algorithms with last-iterate convergence.
> >
> > Actually, their lower bounds are information-theoretic lower bound. In other words, their lower bound apply to any algorithms, regardless of last-iterate or average-iterate convergent algorithms. Since the lower bound in this paper coincides with theirs, the provided lower bound does not provide any new insight.
> >
> > > This tree can be constructed by performing the search
> >
> > I see. This is what I exactly imagined, but is it really easy? For example, the search in no-limit Texas hold’em would require a lot of computation, and doing one traversal seems infeasible (Table 4 of https://poker.cs.ualberta.ca/count_nl_infosets.html)
> >
> > Please feel free to let me know if I miss anything.

---

> > > ### Author Response · Authors · 2025-04-03
> > >
> > > We thank the reviewer's insightful feedback and prompt response! We provide our further responses below.
> > >
> > > **Q1. Lower Bound.**
> > >
> > > We concur that their lower bound coincides with ours. We will revise the parts of the presentation regarding the lower bound in our paper accordingly.
> > >
> > > **Q2. Search on the Game Tree.**
> > >
> > > We thank the reviewer for raising this crucial implementation aspect. Indeed, as the computation complexity of performing searching on the game tree scales as $O(XA)$, on game instances with extremely large infoset-action spaces, performing search on the game trees of such game instances is nearly infeasible in practice. Meanwhile, as the regret minimization lower bound and the sample complexity for learning in IIEFGs with bandit feedback scale with $\Omega(\sqrt{XAK})$ and $\Omega((X A+Y B) / \varepsilon^2)$, the statistical efficiency guarantees for learning on game instances with extremely large infoset-action spaces also seem vacuous, if no function approximation assumptions are further imposed. We will explicitly include this example in our paper and revise our original statement on the LHS of Line 346-349 accordingly.

---

### Official Review · Reviewer_SfND · 2025-03-14

**Overall Recommendation:** 3

**Summary:**

The paper studies two-player zero-sum POMGs, proposes an negentropy-regularization-based algorithm, and establishes the last-iterate convergence. Though the rate seems quite loose, it compares favorably to the rate in a very relevant work Cai et al. [2023] with bandit feedback and in terms of last-iterate guarantees. A worst-case lower bound is also established.

## update after rebuttal

As I have communicated to the authors, I do not think I missed any important contributions/weaknesses in my original assessment and will therefore keep my score unchanged. I support the acceptance of the paper.

**Claims And Evidence:**

The claims are credible and well supported by the mathematical results.

**Essential References Not Discussed:**

I do not find any essential reference missing. Some papers that are worth noting due to their connection to the work in some aspects include:

1) Zeng et al. [2022] and Abe et al. [2024] seem to use regularization in a similar spirit. They add regularization to obtain last-iterate convergence, and decay the weight to ensure that eventually the original problem is solved.

2) Chen et al. [2023] studies two-player zero-sum games, establishes last-iterate convergence, and allows for uncoupled learning.

3) Abe et al. [2023], already referenced in the paper, needs more discussion, especially as Abe et al. [2023] also considers non-full-information feedback.

Abe, K., Sakamoto, M., Ariu, K. and Iwasaki, A., 2024. Boosting Perturbed Gradient Ascent for Last-Iterate Convergence in Games. arXiv preprint arXiv:2410.02388.

Chen, Z., Zhang, K., Mazumdar, E., Ozdaglar, A. and Wierman, A., 2023. A finite-sample analysis of payoff-based independent learning in zero-sum stochastic games. Advances in Neural Information Processing Systems, 36, pp.75826-75883.

Zeng, S., Doan, T. and Romberg, J., 2022. Regularized gradient descent ascent for two-player zero-sum Markov games. Advances in Neural Information Processing Systems, 35, pp.34546-34558.

**Experimental Designs Or Analyses:**

The experiments do not look meaningful. I do not think any conclusion can be drawn. The curves are very close to each other and well within the confidence interval/one standard deviation.

Minor: The paper mentions simulation results in the abstract but does not include them in the main paper.

**Methods And Evaluation Criteria:**

Yes. The convergence is established on the NE gap, a standard measure of optimality.

**Other Comments Or Suggestions:**

1) The virtual transition is an important tool used in the paper. While the authors made an effort to explain this, the discussion is not clear enough for the audience to understand. The authors should consider explaining the virtual transition in a more intuitive manner, using fewer math expressions.

2) I do not agree with the discussion on the advantage of average-iterate convergence on line 055 and below. I understand that last-iterate convergence is usually preferred in the community, but I do not find the reason compelling. Averaged parameters can be easily updated online and requires minimal extra storage. You do not have to maintain the history of all past iterates. I also do not follow the comment that averaging policies is infeasible with non-linear function approximation. What prevents you from averaging the weights of a neural network across iterations? In addition, most existing convergence results are established in the tabular case anyways.

**Other Strengths And Weaknesses:**

Overall, I believe this work makes sufficient contributions to the literature to warrant acceptance. It is in general well-written (modulo the discussion on virtual transition, which I do not follow) and the technical results seem sound.

On the negative side, the sub-optimality of the convergence rate is a concern, and the simulations do not show a clear advantage of the algorithm over the existing ones.

**Questions For Authors:**

No more questions. See above.

**Relation To Broader Scientific Literature:**

N/A.

**Theoretical Claims:**

I only checked the proof of the lower bound, but not the main theorem on the convergence rate of the proposed algorithm. However, I find the arguments reasonable, given the existing literature on the use of negentropy regularizer and Cai et al. [2023].

---

> ### Author Rebuttal · Authors · 2025-03-30
>
> We thank the reviewer for the valuable comments and suggestions. Our response to each question is provided below.
>
> **Q1. "The paper mentions ... not include them in the main paper."**
>
> Thanks for pointing this out. We will be sure to explicitly include more descriptions of the experiments in the main paper body.
>
> **Q2. Additional References.**
>
> Thanks again for noting these additional references! We compare our work with these works as follows:
>
> * [1] establishes algorithms for solving two-player zero-sum MGs with $\widetilde{{O}}(k^{-1 / 3})$ last-iterate convergence. However, they study fully observable MGs while we study partially observable MGs. Further, they require full-information gradient feedback while we only require the bandit feedback. Note that computing the full-information gradient feedback even requires the knowledge of the state transitions (please see Sec. 2.2 of [1]).
> * For the comparisons with [2,3,4], please see our response to Q1 of reviewer T2uG.
>
> * Our work and [5] both study MGs with bandit feedback. However, we remark that they study fully observable MGs while we study partially observable MGs. Further, they require the assumption that the Markov chain can be irreducible and aperiodic for some policy. Therefore, we believe their algorithm is not directly applicable to our problem and our results are not directly comparable to theirs. Besides, our algorithm is OMD-based and can still guarantee a sublinear regret of $\widetilde{{O}}(k^{7 / 8})$ when the opponent is an adversary, while the policy in [5] is updated via constructing approximations of the best response to the opponent's policies and it is not clear whether their algorithm can guarantee a sublinear regret in the presence of a potentially adversarial opponent.
>
> We will include all the above discussions in our revised paper for better clarity.
>
> **Q3. Experiment Results.**
>
> We fully agree that on Lewis Signaling, the performance of our algorithm overlaps with that of the baseline algorithms. However, we believe the experiment results show that our algorithm performs relatively well across all game instances, and though there might be some baseline that performs similarly to our algorithm on some game instances, this baseline algorithm might not be able to converge fast on other game instances, as the last-iterate convergences of all the baseline algorithms are not theoretically guaranteed.
>
> Specifically, as mentioned in Appendix I, on the remaining game instances that are harder to learn (note that $X=3$, $A=3$ on Lewis Signaling, while $X=6$, $A=2$ on Kuhn Poker, $X=468$, $A=3$ on Leduc Poker and $X=12288$, $A=13$ on Liars Dice), only baseline BalancedFTRL converges as fast as our algorithm on Leduc Poker. However, our algorithm converges faster than BalancedFTRL on Liars Dice, and there is clearly a large gap between our algorithm and it on Kuhn Poker during episode $10^5$ to $10^7$. The other baseline with notable performance is BalancedOMD. Nevertheless, we also would like to note that, though BalancedOMD tends to have a similar NE gap at the very end on Kuhn Poker and Leduc Poker, our algorithm converges relatively faster than BalancedOMD up to episode $10^5$ on Kuhn Poker and $10^7$ on Leduc Poker. More importantly, the performance of BalancedOMD is the worst among all the baselines on the hardest game instance Liars Dice, and our algorithm converges clearly faster than BalancedOMD on this instance.
>
> **Q4. "The virtual transition is an important tool used in the paper ..., using fewer math expressions."**
>
> Please see our response to Q2 of reviewer T2uG.
>
> **Q5. "I do not agree with the discussion on ..."**
>
> We agree that if linear function approximation is leveraged, the average policy can be obtained by applying a moving average over the parameters online. Our original statement on the LHS of Line 55-57 might not be appropriate, as we now realize, and we will revise it accordingly. When using nonlinear function approximation, averaging the parameters of nonlinear function approximation is of course possible, but our original statement on the LHS of Line 61-63 intended to indicate that simply averaging the parameters of nonlinear function approximation might not be able to obtain the average policy. To see this, let $f(x)=x^2$, $x_1=1$ and $x_2=5$. Then $f(\frac{x_1+x_2}{2})=9\ne \frac{f(x_1)+f(x_2)}{2}=13$. We will also revise this part for better clarity.
>
> [1] Zeng et al. Regularized Gradient Descent Ascent for Two-Player Zero-Sum Markov Games. NeurIPS, 20.
>
> [2] Abe et al. Last-iterate convergence with full and noisy feedback in two-player zero-sum games. AISTATS, 23.
>
> [3] Abe et al. Adaptively Perturbed Mirror Descent for Learning in Games. ICML, 24.
>
> [4] Abe et al. Boosting Perturbed Gradient Ascent for Last-Iterate Convergence in Games. ICLR, 25.
>
> [5] Chen et al. A finite-sample analysis of payoff-based independent learning in zero-sum stochastic games. NeurIPS, 23.
>
> [6] Lattimore et al. Bandit Algorithms. 20.

---

> > ### Comment · Reviewer_SfND · 2025-04-01
> >
> > I thank the reviewers for the response. I do not think I missed any important contributions/weaknesses in my original assessment and will therefore keep my score unchanged. I support the acceptance of the paper.

---

> > > ### Author Response · Authors · 2025-04-03
> > >
> > > We thank the reviewer for the prompt response and the support of acceptance of our work!

---

### Decision · Program_Chairs · 2025-05-01

**Decision:**

Accept (poster)

**Comment:**

This paper proposes a new algorithm for two-player zero-sum extensive-form games (2p0g) under bandit feedback. In a self-play setting, the last iterate of the strategy generated by the algorithm converges to the Nash equilibrium at a rate of k^{-1/8}. The key innovation lies in replacing the commonly used dilated entropy regularization with negentropy regularization, combined with a novel technique called virtual transition. This is a nice theory result and I recommend for acceptance.